# Identifying Functionally Important Features with End-to-End Sparse Dictionary Learning

**Dan Braun**[*]  **Jordan Taylor**[†]  **Nicholas Goldowsky-Dill**[*]

**Lee Sharkey**[*]

## Abstract

Identifying the features learned by neural networks is a core challenge in mechanistic interpretability. Sparse autoencoders (SAEs), which learn a sparse, overcomplete dictionary that reconstructs a network's internal activations, have been used to identify these features. However, SAEs may learn more about the structure of the dataset than the computational structure of the network. There is therefore only indirect reason to believe that the directions found in these dictionaries are functionally important to the network. We propose end-to-end (e2e) sparse dictionary learning, a method for training SAEs that ensures the features learned are functionally important by minimizing the KL divergence between the output distributions of the original model and the model with SAE activations inserted. Compared to standard SAEs, e2e SAEs offer a Pareto improvement: They explain more network performance, require fewer total features, and require fewer simultaneously active features per datapoint, all with no cost to interpretability. We explore geometric and qualitative differences between e2e SAE features and standard SAE features. E2e dictionary learning brings us closer to methods that can explain network behavior concisely and accurately. We release our library for training e2e SAEs and reproducing our analysis at https://github.com/ApolloResearch/e2e_sae.

## 1 Introduction

Sparse Autoencoders (SAEs) are a popular method in mechanistic interpretability [Sharkey et al., 2022, Cunningham et al., 2023, Bricken et al., 2023]. They have been proposed as a solution to the problem of superposition, the phenomenon by which networks represent more 'features' than they have neurons. 'Features' are directions in neural activation space that are considered to be the basic units of computation in neural networks. SAE dictionary elements (or 'SAE features') are thought to approximate the features used by the network. SAEs are typically trained to reconstruct the activations of an individual layer of a neural network using a sparsely activating, overcomplete set of dictionary elements (directions). It has been shown that this procedure identifies ground truth features in toy models [Sharkey et al., 2022].

However, current SAEs focus on the wrong goal: They are trained to minimize mean squared reconstruction error (MSE) of activations (in addition to minimizing their sparsity penalty). The issue is that the importance of a feature as measured by its effect on MSE may not strongly correlate with how important the feature is for explaining the network's performance. This would not be a problem if the network's activations used a small, finite set of ground truth features – the SAE would simply identify those features, and thus optimizing MSE would have led the SAE to learn the functionally

---

[*]Apollo Research
[†]ML Alignment & Theory Scholars (MATS), University of Queensland

38th Conference on Neural Information Processing Systems (NeurIPS 2024).

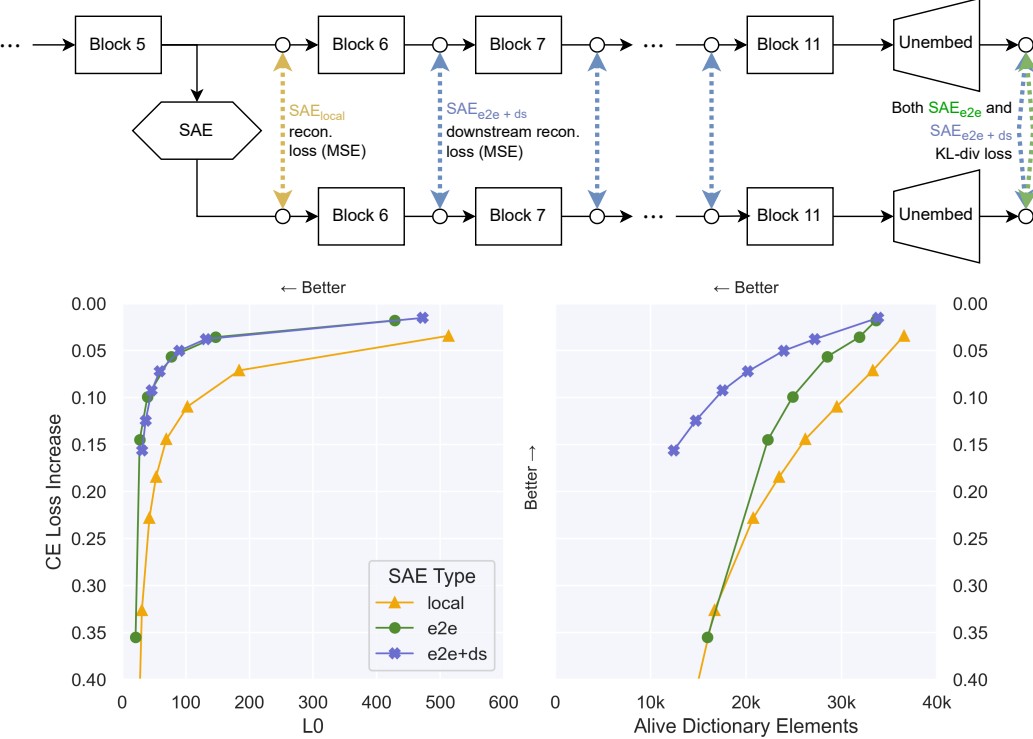

Figure 1: **Top**: Diagram comparing the loss terms used to train each type of SAE. Each arrow is a loss term which compares the activations represented by circles. $SAE_{local}$ uses MSE reconstruction loss between the SAE input and the SAE output. $SAE_{e2e}$ uses KL-divergence on the logits. $SAE_{e2e+ds}$ (end-to-end + downstream reconstruction) uses KL-divergence in addition to the sum of the MSE reconstruction losses at all future layers. All three are additionally trained with a $L_1$ sparsity penalty (not pictured).
**Bottom**: Pareto curves for three different types of GPT2-small layer 6 SAEs as the sparsity coefficient is varied. E2e-SAEs require fewer features per datapoint (i.e. have a lower $L_0$) and fewer features over the entire dataset (i.e. have a low number of alive dictionary elements). GPT2-small has a CE loss of 3.139 over our evaluation set.

important features. In practice, however, Bricken et al. [2023] observed the phenomenon of feature splitting, where increasing dictionary size while increasing sparsity allows SAEs to split a feature into multiple, more specific features, representing smaller and smaller portions of the dataset. In the limit of large dictionary size, it would be possible to represent each individual datapoint as its own dictionary element. Since minimizing MSE does not explicitly prioritize learning features based on how important they are for explaining the network's performance, an SAE may waste much of its fixed capacity on learning less important features. This is perhaps responsible for the observation that, when measuring the causal effects of circuits made from SAE features on network performance, a significant amount is mediated by the reconstruction residual errors (i.e. everything *not* explained by the SAE) and not mediated by SAE features [Marks et al., 2024].

Given these issues, it is therefore natural to ask how we can identify the *functionally important* features used by the network. We say a feature is functionally important if it is important for explaining the network's behavior on the training distribution. If we prioritize learning functionally important features, we should be able to maintain strong performance with fewer features used by the SAE per datapoint as well as fewer overall features.

To optimize SAEs for these properties, we introduce a new training method. We still train SAEs using a sparsity penalty on the feature activations (to reduce the number of features used on each datapoint), but we no longer optimize activation reconstruction. Instead, we replace the original activations with the SAE output (Figure 1) and optimize the KL divergence between the original output logits and the output logits when passing the SAE output through the rest of the network, thus training the SAE end-to-end (e2e). We use $SAE_{e2e}$ to denote an SAE trained with KL divergence and a sparsity penalty.

By contrast, we use SAE$_{local}$ to denote our baseline SAEs, trained only to reconstruct the activations at the current layer with a sparsity penalty.

One risk with this method is that it may be possible for the outputs of SAE$_{e2e}$ to take a different computational pathway through subsequent layers of the network (compared with the original activations) while nevertheless producing a similar output distribution. For example, it might learn a new feature that exploits a particular transformation in a downstream layer that is unused by the regular network or that is used for other purposes. To reduce this likelihood, we also add terms to the loss for the reconstruction error between the original model and the model with the SAE at downstream layers in the network (Figure 1). We use SAE$_{e2e+ds}$ to denote SAEs trained with KL divergence, a sparsity penalty, and downstream reconstruction loss. We use *e2e SAEs* to refer to the family of methods introduced in this work, including both SAE$_{e2e}$ and SAE$_{e2e+ds}$.

Previous work has used the performance explained – measured by cross-entropy loss difference when replacing the original activations with SAE outputs – as a measure of SAE quality [Cunningham et al., 2023, Bricken et al., 2023, Bloom, 2024]. It's reasonable to ask whether our approach runs afoul of Goodhart's law ("*When a measure becomes a target, it ceases to be a good measure*"). We contend that mechanistic interpretability should prefer explanations of networks (and the components of those explanations, such as features) that explain more network performance over other explanations. Therefore, optimizing directly for quantitative proxies of performance explained (such as CE loss difference, KL divergence, and downstream reconstruction error) is preferred.

We train each SAE type on language models (GPT2-small [Radford et al., 2019] and Tinystories-1M [Eldan and Li, 2023]), and present three key findings:

1. For the same level of performance explained, SAE$_{local}$ requires activating more than twice as many features per datapoint compared to SAE$_{e2e+ds}$ and SAE$_{e2e}$ (Section 3.1).

2. SAE$_{e2e+ds}$ performs equally well as SAE$_{e2e}$ in terms of the number of features activated per datapoint (Section 3.1), yet its activations take pathways through the network that are much more similar to SAE$_{local}$ (Sections 3.2, 3.3).

3. SAE$_{local}$ requires more features in total over the dataset to explain the same amount of network performance compared with SAE$_{e2e}$ and SAE$_{e2e+ds}$ (Section 3.1).

These findings suggest that e2e SAEs are more efficient in capturing the essential features that contribute to the network's performance. Moreover, our automated-interpretability and qualitative analyses reveal that SAE$_{e2e+ds}$ features are at least as interpretable as SAE$_{local}$ features, demonstrating that the improvements in efficiency do not come at the cost of interpretability (Section 3.4). These gains nevertheless come at the cost of longer wall-clock time to train (Appendix H).

As a supplementary investigation, we tested e2e SAEs on a set of subject-verb agreement tasks from Finlayson et al. [2021]. These tests had inconclusive results (Appendix I), indicating that further work is needed to identify which downstream tasks would benefit most from e2e SAEs.

In addition to this article, we also provide: A library for training all SAE types presented in this article (https://github.com/ApolloResearch/e2e_sae); a Weights and Biases [Biewald, 2020] report that links to training metrics for all runs (https://api.wandb.ai/links/sparsify/evnqx8t6); a Neuronpedia [Lin and Bloom, 2023] page for interacting with the features in a subset of SAEs (those presented in Tables 2 and 3) (https://www.neuronpedia.org/gpt2sm-apollojt) as well as a repository for downloading these SAEs directly (https://huggingface.co/apollo-research/e2e-saes-gpt2).

## 2 Training end-to-end SAEs

Our experiments train SAEs using three kinds of loss function (Figure 1), which we evaluate according to several metrics (Section 2.5):

1. $L_{local}$ trains SAEs to reconstruct activations at a particular layer (Section 2.2);

2. $L_{e2e}$ trains SAEs to learn functionally important features (Section 2.3);

3. $L_{e2e+downstream}$ trains SAEs to learn functionally important features that optimize for faithfulness to the activations of the original network at subsequent layers (Section 2.4).

## 2.1 Formulation

Suppose we have a feedforward neural network (such as a decoder-only Transformer [Radford et al., 2018]) with $L$ layers and vectors of hidden activations $a^{(l)}$:

$$a^{(0)}(x) = x$$
$$a^{(l)}(x) = f^{(l)}(a^{(l-1)}(x)), \text{ for } l = 1, \ldots, L-1$$
$$y = \text{softmax}\left(f^{(L)}(a^{(L-1)}(x))\right).$$

We use SAEs that consist of an encoder network (an affine transformation followed by a ReLU activation function) and a dictionary of unit norm features, represented as a matrix $D$, with associated bias vector $b_d$. The encoder takes as input network activations from a particular layer $l$. The architecture we use is:

$$\text{Enc}\left(a^{(l)}(x)\right) = \text{ReLU}\left(W_e a^{(l)}(x) + b_e\right)$$
$$\text{SAE}\left(a^{(l)}(x)\right) = D^{\top}\text{Enc}\left(a^{(l)}(x)\right) + b_d,$$

where the dictionary $D$ and encoder weights $W_e$ are both (`N_dict_elements` $\times$ `d_hidden`) matrices, $b_e$ is a `N_dict_elements`-dimensional vector, while $b_d$ and $a^{(l)}(x)$ are `d_hidden`-dimensional vectors.

## 2.2 Baseline: Local SAE training loss ($L_{\text{local}}$)

The standard, baseline method for training SAEs is $\text{SAE}_{\text{local}}$ training, where the output of the SAE is trained to reconstruct its input using a mean squared error loss with a sparsity penalty on the encoder activations (here an L1 loss):

$$L_{\text{local}} = L_{\text{reconstruction}} + L_{\text{sparsity}} = ||a^{(l)}(x) - \text{SAE}_{\text{local}}(a^{(l)}(x))||_2^2 + \phi||\text{Enc}(a^{(l)}(x))||_1.$$

$\phi = \frac{\lambda}{\dim(a^{(l)})}$ is a sparsity coefficient $\lambda$ scaled by the size of the input to the SAE (see Appendix D for details on hyperparameters).

## 2.3 Method 1: End-to-end SAE training loss ($L_{\text{e2e}}$)

For $\text{SAE}_{\text{e2e}}$, we do not train the SAE to reconstruct activations. Instead, we replace the model activations with the output of the SAE and pass them forward through the rest of the network:

$$\hat{a}^{(l)}(x) = \text{SAE}_{\text{e2e}}(a^{(l)}(x))$$
$$\hat{a}^{(k)}(x) = f^{(k)}(\hat{a}^{(l)}(x)) \text{ for } k = l, \ldots, L-1$$
$$\hat{y} = \text{softmax}\left(f^{(L)}(\hat{a}^{(L-1)}(x))\right)$$

We train the SAE by penalizing the KL divergence between the logits produced by the model with the SAE activations and the original model:

$$L_{\text{e2e}} = L_{\text{KL}} + L_{\text{sparsity}} = KL(\hat{y}, y) + \phi||\text{Enc}(a^{(l)}(x))||_1$$

Importantly, we freeze the parameters of the model, so that only the SAE is trained. This contrasts with Tamkin et al. [2023], who train the model parameters in addition to training a 'codebook' (which is similar to a dictionary).

## 2.4 Method 2: End-to-end with downstream layer reconstruction SAE training loss ($L_{\text{e2e+downstream}}$)

A reasonable concern with the $L_{\text{e2e}}$ is that the model with the SAE inserted may compute the output using an importantly different pathway through the network, even though we've frozen the original model's parameters and trained the SAE to replicate the original model's output distribution. To counteract this possibility, we also compare an additional loss: The end-to-end with downstream

reconstruction training loss ($L_{\text{e2e+downstream}}$) additionally minimizes the mean squared error between the activations of the new model at downstream layers and the activations of the original model:

$$L_{\text{e2e+downstream}} = L_{\text{KL}} + L_{\text{sparsity}} + L_{\text{downstream}}$$

$$= KL(\hat{y}, y) + \phi||\text{Enc}(a^{(l)})||_1 + \frac{\beta_l}{L-l} \sum_{k=l+1}^{L-1} ||\hat{a}^{(k)}(x) - a^{(k)}(x)||_2^2 \quad (1)$$

where $\beta_l$ is a hyperparameter that controls the downstream reconstruction loss term (Appendix D).

$L_{\text{e2e+downstream}}$ thus has the desirable properties of 1) incentivizing the SAE outputs to lead to similar computations in downstream layers in the model and 2) allowing the SAE to "clear out" some of the non-functional features by not training on a reconstruction error at the layer with the SAE. Note, however, the inclusion of the intermediate reconstruction terms means that $L_{\text{e2e+downstream}}$ may encourage the SAE to learn features that are less functionally important.

### 2.5 Experimental metrics

We record several key metrics for each trained SAE:

1. **Cross-entropy loss increase** between the original model and the model with SAE: We measure the increase in cross-entropy (CE) loss caused by using activations from the inserted SAE rather than the original model activations on an evaluation set. We sometimes refer to this as 'amount of performance explained', where a low CE loss increase means more performance explained. *All other things being equal, a better SAE recovers more of the original model's performance.*

2. **$L_0$**: How many SAE features activate on average for each datapoint. *All other things being equal, a better SAE needs fewer features to explain the performance of the model on a given datapoint.*

3. **Number of alive dictionary elements**: The number of features in training that have not 'died' (which we define to mean that they have not activated over a set of 500k tokens of data). *All other things being equal, a better SAE needs a smaller number of alive features to explain the performance of model over the dataset.*

We also record the **reconstruction loss at downstream layers**. This is the mean squared error between the activations of the original model and the model with the SAE at all layers following the insertion of the SAE (i.e. downstream layers). If reconstruction loss at downstream layers is low, then the activations take a similar pathway through the network as in the original model. This minimizes the risk that the SAEs are learning features that take different computational pathways through the downstream layers compared to the original model. Finally, following Bills et al. [2023], we perform **automated-interpretability scoring** and qualitative analysis on a subset of the SAEs, to verify that improved quantitative metrics does not sacrifice the interpretability of the learned features.

We show results for experiments performed on GPT2-small's residual stream before attention layer 6.[3] Results for layers 2, 6, and 10 of GPT2-small and some runs on a model trained on the TinyStories dataset [Eldan and Li, 2023] can be found in Appendices A.1 and A.2, respectively. They are qualitatively similar to those presented in the main text. For our GPT2-small experiments, we train SAEs with each type of loss function on 400k samples of context size 1024 from the Open Web Text dataset [Gokaslan and Cohen, 2019] over a range of sparsity coefficients $\lambda$. Our dictionary is fixed at 60 times the size of the residual stream (i.e. $60 \times 768 = 46080$ initial dictionary elements). Hyperparameters, along with sweeps over dictionary size and number of training examples, are shown in Appendices D and E, respectively.

## 3 Results

### 3.1 End-to-end SAEs are a Pareto improvement over local SAEs

We compare the trained SAEs according to CE loss increase, $L_0$, and number of alive dictionary elements. The learning rates for each SAE type were selected to be Pareto-optimal according to their

---

[3]We use zero-indexed layer numbers throughout this article

$L_0$ vs CE loss increase curves.[4] Each experiment uses a range of sparsity coefficients $\lambda$. In Figure 1, we see that both $SAE_{e2e}$ and $SAE_{e2e+ds}$ achieve better CE loss increase for a given $L_0$ or for a given number of alive dictionary elements. This means they need fewer features to explain the same amount of network performance for a given datapoint or for the dataset as a whole, respectively. For similar results at other layers see Appendix A.1.

This difference is large: For a given $L_0$, both $SAE_{e2e}$ and $SAE_{e2e+ds}$ have a CE loss increase that is less than $45\%$ of the CE loss increase of $SAE_{local}$.[5] $SAE_{local}$ must therefore be learning features that are not maximally important for explaining network performance.

This improved performance comes at the expense of increased compute costs (2-3.5 times longer runtime, see Appendix H). We test to see if additional compute improves our $SAE_{local}$ baseline in Appendix E. We find neither increasing dictionary size from $60 * 768$ to $100 * 768$ nor increasing training samples from 400k to 800k noticeably improves the Pareto frontier, implying that our e2e SAEs maintain their advantage even when compared against $SAE_{local}$ dictionaries trained with more compute.

For comparability, our subsequent analyses focus on 3 particular SAEs that have approximately equivalent CE loss increases (Table 1).

Table 1: Three SAEs from layer 6 with similar CE loss increases are analyzed in detail.

| SAE Type | $\lambda$ (Sparsity Coeff) | $L_0$ | Alive Elements | CE Loss Increase |
|---|---|---|---|---|
| Local | 4.0 | 69.4 | 26k | **0.145** |
| End-to-end | 3.0 | 27.5 | 22k | **0.144** |
| E2e + Downstream | 50.0 | 36.8 | 15k | **0.125** |

## 3.2 End-to-end SAEs have worse reconstruction loss at each layer despite similar output distributions

Even though $SAE_{e2e}$s explain more performance per feature than $SAE_{local}$s, they have much worse reconstruction error of the original activations at each subsequent layer (Figure 2). This indicates that the activations following the insertion of $SAE_{e2e}$ take a different path through the network than in the original model, and therefore potentially permit the model to achieve its performance using different computations from the original model. This possibility motivated the training of $SAE_{e2e+ds}$s.

In later layers, the reconstruction errors of $SAE_{local}$ and $SAE_{e2e+ds}$ are extremely similar (Figure 2). $SAE_{e2e+ds}$ therefore has the desirable properties of both learning features that explain approximately as much network performance as $SAE_{e2e}$ (Figure 1) while having reconstruction errors that are much closer to $SAE_{local}$. There remains a difference in reconstruction at layer 6 between $SAE_{e2e+ds}$ and $SAE_{local}$. This is not surprising given that $SAE_{e2e+ds}$ is not trained with a reconstruction loss at this layer. In Appendix B, we examine how much of this difference is explained by feature scaling. In Appendix G.3, we find a specific example of a direction with low functional importance that is faithfully represented in $SAE_{local}$ but not $SAE_{e2e+ds}$.

## 3.3 Differences in feature geometries between SAE types

### 3.3.1 End-to-end SAEs have more orthogonal features than $SAE_{local}$

Bricken et al. [2023] observed 'feature splitting', where a locally trained SAEs learns a cluster of features which represent similar categories of inputs and have dictionary elements pointing in similar directions. A key question is to what extent these subtle distinctions are functionally important for the

---

[4]We show in Appendix C that it is possible to reduce the number of alive dictionary elements for any SAE type by increasing the learning rate. This has minimal cost according to $L_0$ vs CE loss increase Pareto-optimality up to some limit.

[5]Measured using linear interpolation over a range of $L_0 \in (50, 300)$. This range was chosen based on two criteria: (1) $L_0$ should be significantly smaller than the residual stream size for the SAE to be effective (we conservatively chose 300 compared to the residual stream size of 768), and (2) the CE loss should not start to increase dramatically, which occurs at approximately $L_0 = 50$ (Figure 1).

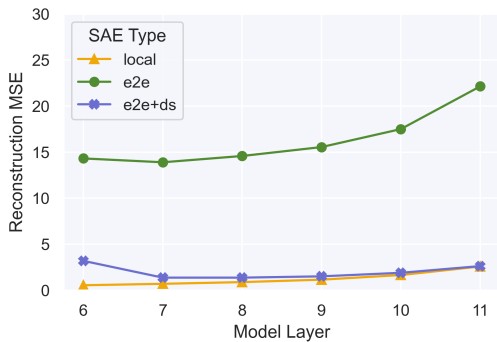

Figure 2: Reconstruction mean squared error (MSE) at later layers for our set of GPT2-small layer 6 SAEs with similar CE loss increases (Table 1). SAE$_{local}$ is trained to minimize MSE at layer 6, SAE$_{e2e}$ was trained to match the output probability distribution, SAE$_{e2e+ds}$ was trained to match the output probability distribution *and* minimize MSE in all downstream layers.

network's predictions, or if they are only helpful for reconstructing functionally unimportant patterns in the data.

We have already seen that SAE$_{e2e}$ and SAE$_{e2e+ds}$ learn smaller dictionaries compared with SAE$_{local}$ for a given level of performance explained (Figure 1). In this section, we explore if this is due to less feature splitting. We measure the cosine similarities between each SAE dictionary feature and next-closest feature in the same dictionary. While this does not account for potential semantic differences between directions with high cosine similarities, it serves as a useful proxy for feature splitting, since split features tend to be highly similar directions [Bricken et al., 2023]. We find that SAE$_{local}$ has features that are more tightly clustered, suggesting higher feature splitting (Figure 3a). Compared to SAE$_{e2e+ds}$ the mean cosine similarity is $0.04$ higher (bootstrapped 95% CI $[0.037 - 0.043]$); compared to SAE$_{e2e}$ the difference is $0.166$ (95% CI $[0.163 - 0.168]$). We measure this for all runs in our Pareto frontiers and find that this difference is not explained by SAE$_{local}$ having more alive dictionary elements than e2e SAEs (Appendix A.5).

### 3.3.2 SAE$_{e2e}$ features are not robust across random seeds, but SAE$_{e2e+ds}$ and SAE$_{local}$ are

We find that SAE$_{local}$s trained with one seed learn similar features as SAE$_{local}$s trained with a different seed (Figure 3b). The same is true for two SAE$_{e2e+ds}$s. However, features learned by SAE$_{e2e}$ are quite different for different seeds. This suggests there are many different sets of SAE$_{e2e}$ features that achieve the same output distribution, despite taking different paths through the network.

### 3.3.3 SAE$_{e2e}$ and SAE$_{e2e+ds}$ features do not always align with SAE$_{local}$ features

The cosine similarity plots between SAE$_{e2e}$ and SAE$_{local}$ (Figure 3c top) reveal that the average similarity between the most similar features is low, and includes a group of features that are very dissimilar. SAE$_{e2e+ds}$ learns features that are much more similar to SAE$_{local}$, although the cosine similarity plot is bimodal, suggesting that SAE$_{e2e+ds}$ learns a set of directions that very different to those identified by SAE$_{local}$ (Figure 3c bottom).

It is encouraging that SAE$_{local}$ and SAE$_{e2e+ds}$ features are somewhat similar, since this indicates that SAE$_{local}$s may serve as good initializations for training SAE$_{e2e+ds}$s, reducing training time.

### 3.4 Interpretability of learned directions

Using the `automated-interpretability` library [Lin, 2024] (an adaptation of Bills et al. [2023]), we generate automated explanations of our SAE features by prompting *gpt-4-turbo-2024-04-09* [OpenAI et al., 2024] with five max-activating examples for each feature, before generating "interpretability scores" by tasking *gpt-3.5-turbo* to use that explanation to predict the SAE feature's true activations on a random sample of 20 max-activating examples. For each SAE we generate automated-interpretabilty scores for a random sample of features ($n = 198$ to $201$ per SAE). We then measure the difference between average interpretability scores. This interpretability score is an

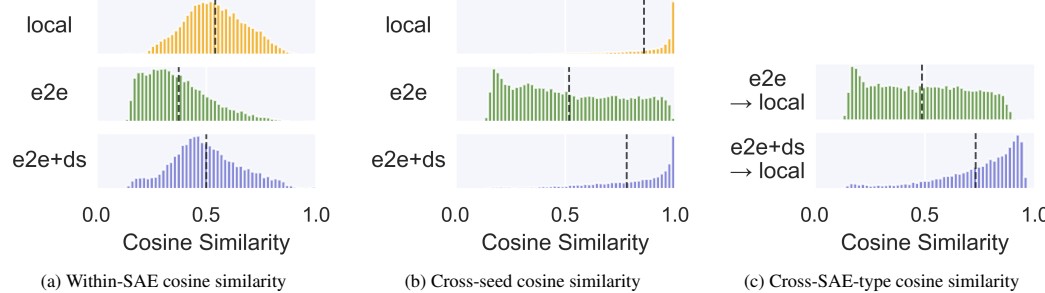

|  (a) Within-SAE cosine similarity | (b) Cross-seed cosine similarity | (c) Cross-SAE-type cosine similarity |

Figure 3: Geometric comparisons for our set of GPT2-small layer 6 SAEs with similar CE loss increases (Table 1). For each dictionary element, we find the max cosine similarity between itself and all other dictionary elements. In 3a we compare to others directions in the same SAE, in 3b to directions in an SAE of the same type trained with a different random seed, in 3c to directions in the $SAE_{local}$ with similar CE loss increase.

imperfect metric of interpretability, but it serves as an unbiased verification and is therefore useful for ensuring that we are not trading better training losses for significantly less interpretable features.

For pairs of SAEs with similar $L_0$ (listed in Table 3), we find no difference between the average interpretability scores of $SAE_{local}$ and $SAE_{e2e+ds}$. If we repeat the analysis for pairs with similar CE loss increases, we find the $SAE_{e2e+ds}$ features to be more interpretable than $SAE_{local}$ features in Layers 2 ($p = 0.0053$) and 6 ($p = 0.0005$) but no significant difference in layer 10. For additional automated-interpretability analysis, see Appendix A.7.

We also provide some qualitative, human-generated interpretations of some groups of features for different SAE types in Appendix G. Features from the SAEs in Table 2 and Table 3 can be viewed interactively at `https://www.neuronpedia.org/gpt2sm-apollojt`.

# 4 Related work

## 4.1 Using sparse autoencoders and sparse coding in mechanistic interpretability

When Elhage et al. [2022] identified superposition as a key bottleneck to progress in mechanistic interpretability, the field found a promising scalable solution in SAEs [Sharkey et al., 2022]. SAEs have since been used to interpret language models [Cunningham et al., 2023, Bricken et al., 2023, Bloom, 2024] and have been used to improve performance of classifiers on downstream tasks [Marks et al., 2024]. Earlier work by Yun et al. [2021] concatenated together the residual stream of a language model and used sparse coding to identify an undercomplete set of sparse 'factors' that spanned multiple layers. This echoes even earlier work that applied sparse coding to word embeddings and found sparse linear structure [Faruqui et al., 2015, Subramanian et al., 2017, Arora et al., 2018]. Similar to our work is Tamkin et al. [2023], who trained sparse feature codebooks, which are similar to SAEs, and trained them end-to-end. However, to achieve adequate performance, they needed to train the model parameters alongside the sparse codebooks. Here, we only trained the SAEs and left the interpreted model unchanged.

## 4.2 Identifying problems with and improving sparse autoencoders

Although useful for mechanistic interpretability, current SAE approaches have several shortcomings. One issue is the functional importance of features, which we have aimed to address here. Some work has noted problems with SAEs, including Anders and Bloom [2024], who found that SAE features trained on a language model with a given context length failed to generalize to activations collected from activations in longer contexts. Other work has addressed 'feature suppression' [Wright and Sharkey, 2024], also known as 'shrinkage' [Jermyn et al., 2024], where SAE feature activations systematically undershoot the 'true' activation value because of the sparsity penalty. While Wright and Sharkey [2024] approached this problem using finetuning after SAE training, Jermyn et al. [2024] and Riggs and Brinkmann [2024] explored alternative sparsity penalties during training that aimed to reduce feature suppression (with mixed success). Farrell [2024], taking an approach similar to Jermyn et al. [2024], has explored different sparsity penalties, though here not to address shrinkage,

but instead to optimize for other metrics of SAE quality. Rajamanoharan et al. [2024] introduce Gated SAEs, an architectural variation for the encoder which both addresses shrinkage and improves on the Pareto frontier of $L_0$ vs CE loss increase.

### 4.3 Methods for evaluating the quality of trained SAEs

One of the main challenges in using SAEs for mechanistic interpretability is that there is no known 'ground truth' against which to benchmark the features learned by SAEs. Prior to our work, several metrics have been used, including: Comparison with ground truth features in toy data; activation reconstruction loss; $L_1$ loss; number of alive dictionary elements; similarity of SAE features across different seeds and dictionary sizes [Sharkey et al., 2022]; $L_0$; KL divergence (between the output distributions of the original model and the model with SAE activations) upon causal interventions on the SAE features [Cunningham et al., 2023]; reconstructed negative log likelihood of the model with SAE activations inserted [Cunningham et al., 2023, Bricken et al., 2023]; feature interpretability Cunningham et al. [2023] (as measured by automatic interpretability methods [Bills et al., 2023]); and task-specific comparisons [Makelov et al., 2024]. In our work, we use (1) $L_0$, (2) number of alive dictionary elements, (3) the average KL divergence between the output distribution of the original model and the model with SAE activations, and (4) the reconstruction error of activations in layers that follow the layer where we replace the original model's activations with the SAE activations.

### 4.4 Methods for identifying the functional importance of sparse features

In our work, we optimize for functional importance directly, but previous work measured functional importance *post hoc* using different approaches. Cunningham et al. [2023] used activation patching [Vig et al., 2020], a form of causal mediation analysis, where they intervened on feature activations and found the output distribution was more sensitive (had higher KL divergence with the original model's distribution) in the direction of SAE features than other directions, such as PCA directions. With the same motivation, Marks et al. [2024] use a similar, approximate, but more efficient, method of causal mediation analysis [Nanda, 2022, Sundararajan et al., 2017]. Unlike our work, these works use the measures of functional importance to construct circuits of sparse features. Bricken et al. [2023] used logit attribution, measuring the effect the feature has on the output logits.

## 5 Conclusion

In this work, we introduce end-to-end dictionary learning as a method for training SAEs to identify functionally important features in neural networks. By optimizing SAEs to minimize the KL divergence between the output distributions of the original model and the model with SAE activations inserted, we demonstrate that e2e SAEs learn features that better explain network performance compared to the standard locally trained SAEs.

Our experiments on GPT2-small and Tinystories-1M reveal several key findings. First, for a given level of performance explained, e2e SAEs require activating significantly fewer features per datapoint and fewer total features over the entire dataset. Second, $SAE_{e2e+ds}$, which has additional loss terms for the reconstruction errors at downstream layers in the model, achieves a similar performance explained to $SAE_{e2e}$ while maintaining activations that follow similar pathways through later layers compared to the original model. Third, the improved efficiency of e2e SAEs does not come at the cost of interpretability, as measured by automated-interpretability scores and qualitative analysis.

These results suggest that standard, locally trained SAEs are capturing information about dataset structure that is not maximally useful for explaining the algorithm implemented by the network. By directly optimizing for functional importance, e2e SAEs offer a more targeted approach to identifying the essential features that contribute to a network's performance.

## 6 Impact statement

This article proposes an improvement to methods used in mechanistic interpretability. Mechanistic interpretability, and interpretability broadly, promises to let us understand the inner workings of neural networks. This may be useful for debugging and improving issues with neural networks. For instance, it may enable the evaluation of a model's fairness or bias. Interpretability may relatedly be

useful for improving the trust-worthiness of AI systems, potentially enabling AI's use in certain high stakes settings, such as healthcare, finance, and justice. However, increasing the trust-worthiness of AI systems may be dual use in that may also enable its use in settings such as military applications.

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

# A Additional results on other layers and models

## A.1 Pareto curves for SAEs at other layers

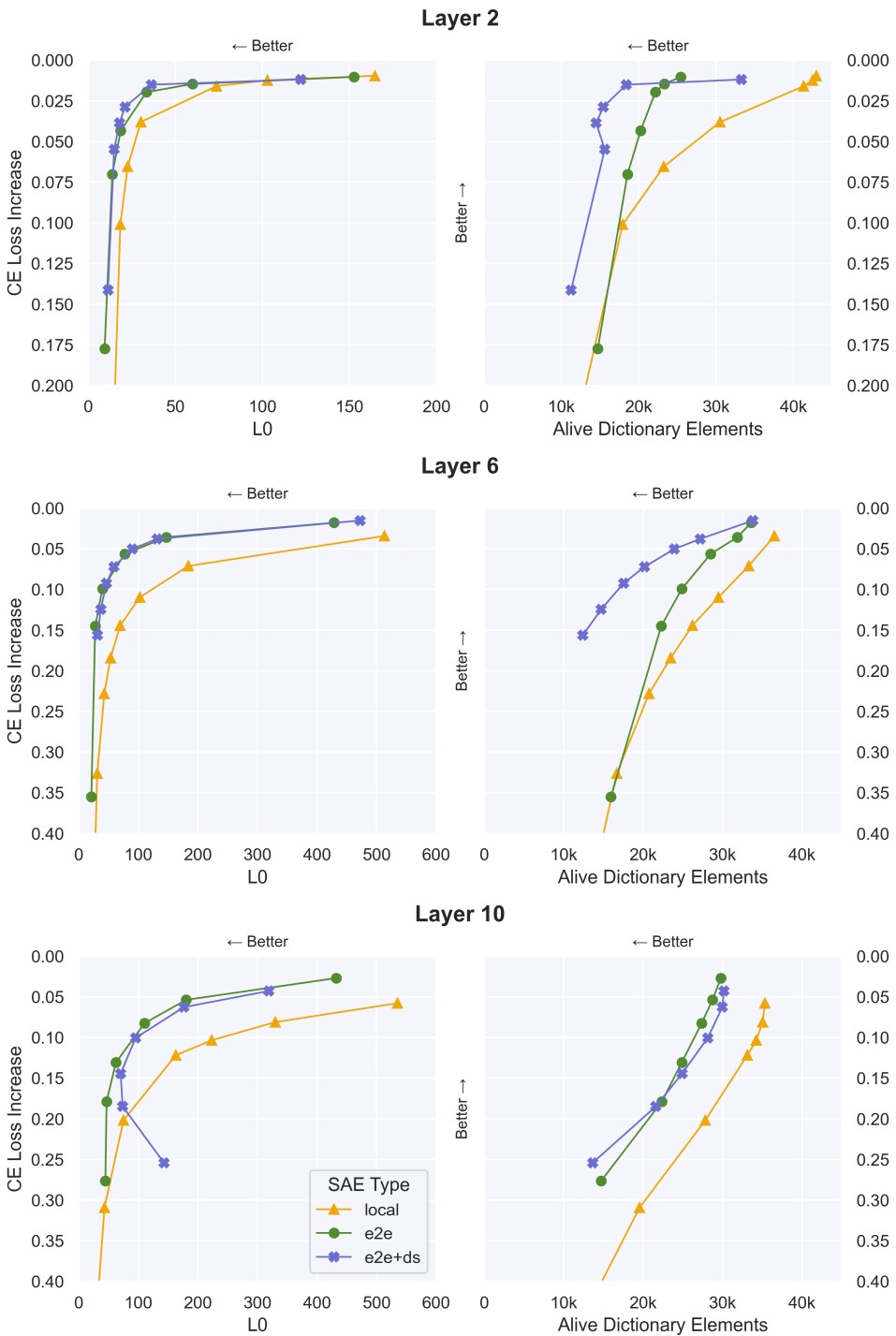

Figure 4: Performance of all SAE types on GPT2-small's residual stream at layers 2, 6 and 10. GPT2-small has a CE loss of 3.139 over our evaluation set.

## A.2 Pareto curves for TinyStories-1M

We also tested our methods on Tinystories-1M, a $1M$ parameter model trained on short, simple stories [Eldan and Li, 2023]. Figure 5 shows our key results generalising to the residual stream halfway through the model (before the $5^{\text{th}}$ of 8 layers).

Note that most of our Tinystories-1M runs were for $\text{SAE}_{\text{local}}$ and $\text{SAE}_{\text{e2e}}$, and we did not perform several of the analyses that we performed for GPT2-small elsewhere in this report. But the clear improvement in $L_0$ and alive_dict_elements vs CE loss increase was apparent for $\text{SAE}_{\text{e2e}}$ vs $\text{SAE}_{\text{local}}$. More results can be found at `https://api.wandb.ai/links/sparsify/yk5etolk`. Future work would test that these results hold on more models of different sizes and architectures, as well as on SAEs trained not just on the residual stream.

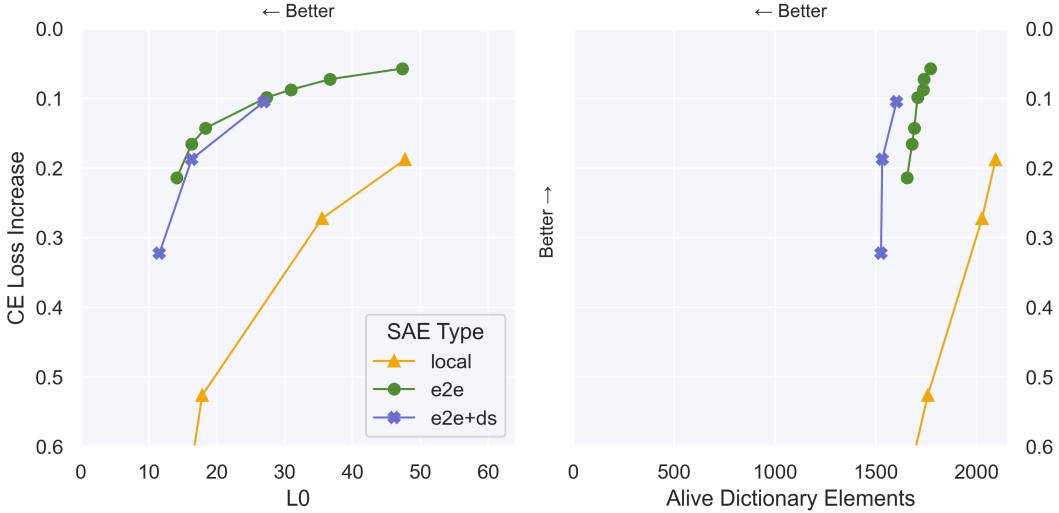

Figure 5: Tinystories-1M runs comparing $\text{SAE}_{\text{local}}$, $\text{SAE}_{\text{e2e}}$ and $\text{SAE}_{\text{e2e+ds}}$ on the residual stream before the $5^{\text{th}}$ of 8 layers. Tinystories-1M has a CE loss of 2.306 over our evaluation set.

## A.3 Comparison of runs with similar $L_0$, CE loss increase, or number of alive dictionary elements

Table 2: Comparison of runs with similar CE loss increase for each layer. $\lambda$ represents the sparsity coefficient and GradNorm is the mean norm of all SAE weight gradients measured from 10k training samples onwards.

| Layer | SAE Type | $\lambda$ | $L_0$ | AliveElements | GradNorm | **CEIncrease** |
|-------|----------|-----------|-------|---------------|----------|----------------|
| 2 | local | 0.8 | 73.5 | 41k | 0.04 | **0.016** |
|   | e2e | 0.5 | 33.4 | 22k | 0.24 | **0.020** |
|   | e2e+ds | 10 | 36.2 | 18k | 1.55 | **0.015** |
| 6 | local | 4 | 69.4 | 26k | 0.12 | **0.144** |
|   | e2e | 3 | 27.5 | 22k | 0.59 | **0.145** |
|   | e2e+ds | 50 | 36.8 | 15k | 3.27 | **0.124** |
| 10 | local | 6 | 162.7 | 33k | 0.40 | **0.122** |
|    | e2e | 1.5 | 62.3 | 25k | 0.64 | **0.131** |
|    | e2e+ds | 1.75 | 70.2 | 25k | 0.24 | **0.144** |

Table 3: Comparison of runs with similar $L_0$ for each layer. $\lambda$ represents the sparsity coefficient and GradNorm is the mean norm of all SAE weight gradients measured from 10k training samples onwards.

| Layer | SAE Type | $\lambda$ | $\mathbf{L_0}$ | Alive Elements | Grad Norm | CE Loss Increase |
|---|---|---|---|---|---|---|
| 2 | local | 4 | **18.4** | 18k | 0.04 | 0.101 |
| | e2e | 1.5 | **18.6** | 20k | 0.31 | 0.043 |
| | e2e+ds | 35 | **17.9** | 15k | 2.24 | 0.039 |
| 6 | local | 6 | **42.8** | 21k | 0.13 | 0.228 |
| | e2e | 1.5 | **39.7** | 25k | 0.42 | 0.099 |
| | e2e+ds | 50 | **36.8** | 15k | 3.27 | 0.124 |
| 10 | local | 10 | **74.9** | 28k | 0.37 | 0.202 |
| | e2e | 1.5 | **62.3** | 25k | 0.64 | 0.131 |
| | e2e+ds | 1.75 | **70.2** | 25k | 0.24 | 0.144 |

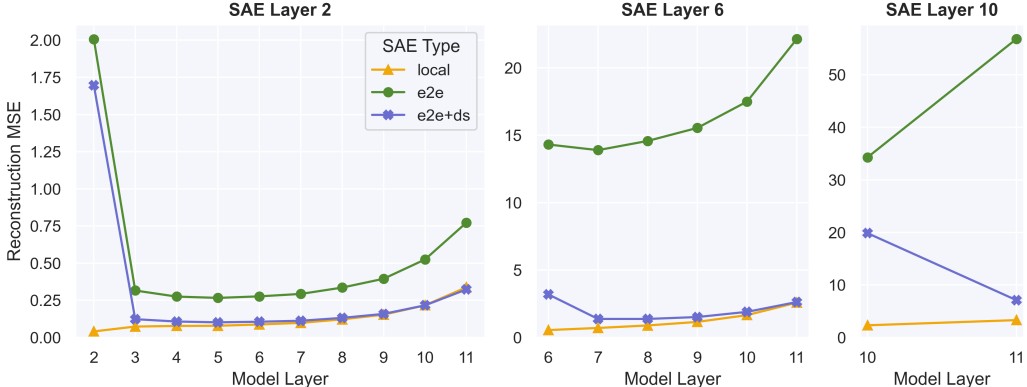

Figure 6: Reconstruction mean squared error (MSE) at later layers for our three SAEs with similar CE loss increase for layers 2, 6, and 10.

## A.4 Downstream MSE for all layers

Figure 6 shows that layers 2 and 10 also have the property that $SAE_{e2e+ds}$ has a very similar reconstruction loss to $SAE_{local}$ at downstreams layers, and $SAE_{e2e}$ has a much higher reconstruction loss.

## A.5 Feature splitting geometry

In Section 3.3.1 we showed that at layer 6, $SAE_{local}$ is less orthogonal than $SAE_{e2e}$ and $SAE_{e2e+ds}$, indicating a higher level of feature splitting. In Figure 7 we extend the analysis to runs on other layers.

In almost all cases we find that $SAE_{e2e}$ contains the most orthogonal dictionaries, followed by $SAE_{e2e+ds}$ and then $SAE_{local}$. Perhaps surprisingly, as the number of alive dictionary elements decrease for each SAE type, we see an increase in the mean of the within-SAE similarities, indicating less feature splitting. One hypothesis for this result is that the the orthogonality of the dictionary depends much more on the output performance (as measured by CE loss difference) or sparsity (as measured by $L_0$) of the model with the SAE than on the number of alive dictionary elements, though further analysis is needed.

## A.6 Cross-type similarity at other layers

In Section 3.3.2 we show that downstream and local SAEs have more similar decoder directions than e2e and local SAEs. In Figure 8 we show this is true for layers 2, 6, and 10.

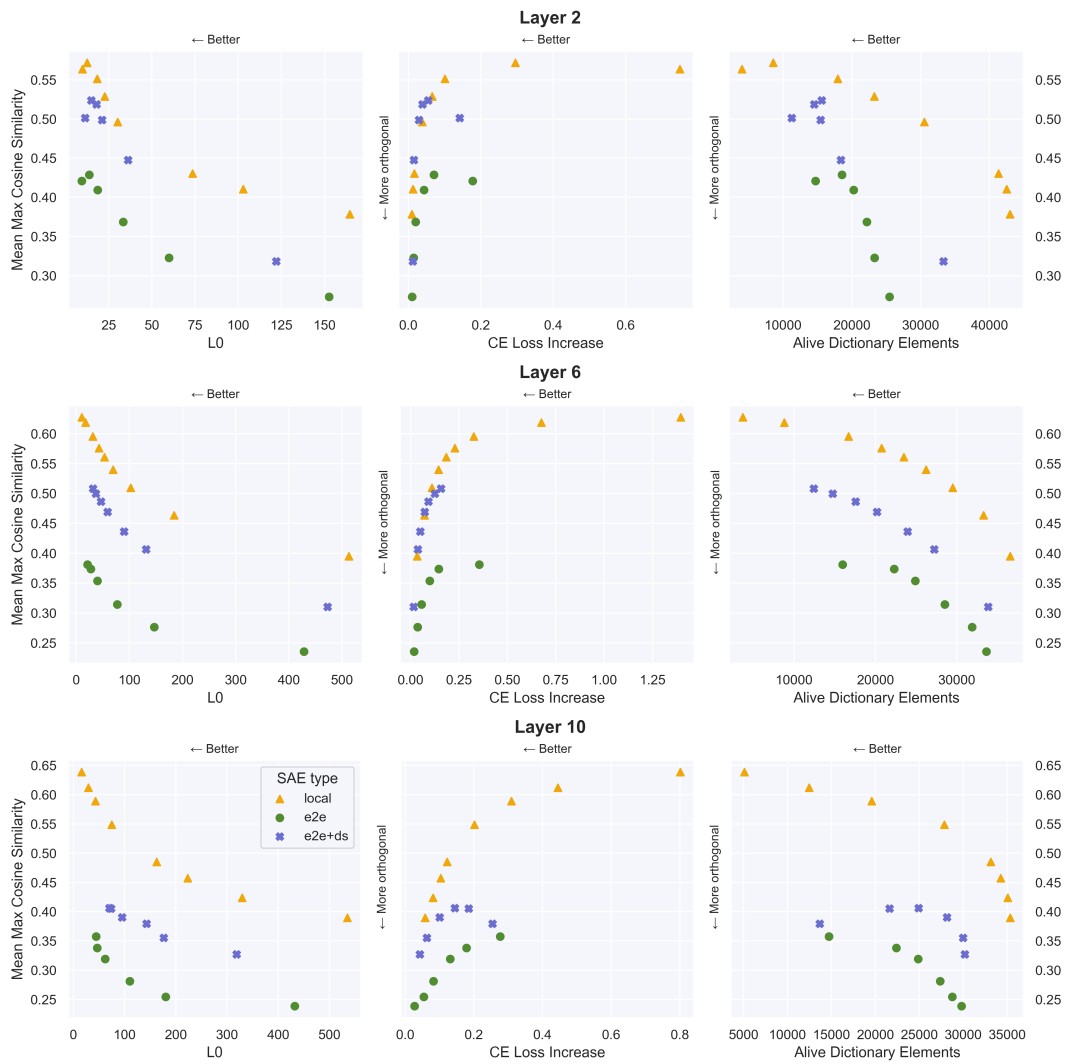

Figure 7: Mean over all SAE dictionary elements of the cosine similarity to the next-closest element in the same dictionary. Plotted against $L_0$, CE loss increase, and number of alive dictionary elements for all SAE types on runs with a variety of sparsity coefficients for GPT2-small

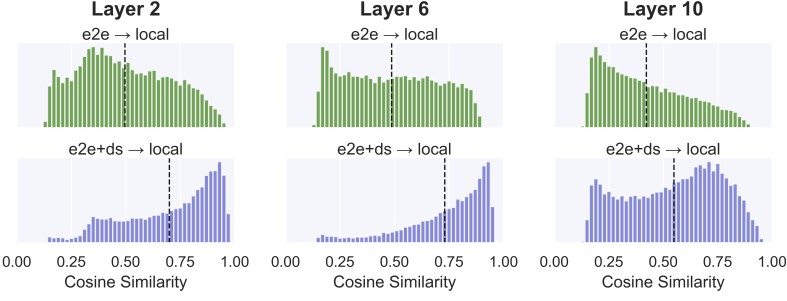

Figure 8: For runs with similar CE loss increase in layers 2, 6, 10, for each $SAE_{e2e}$ and $SAE_{e2e+ds}$ dictionary direction, we take the max cosine similarity over all $SAE_{local}$ directions.

### A.7 Auto-interpretability

In Section 3.4 we claim that when comparing auto-interpretability scores we find no difference between pairs of similar $L_0$, but do find $SAE_{e2e+ds}$ is more interpretable than $SAE_{local}$ in layers 2 and 6. These results are presented in more detail in Figure 9 and Table 4.

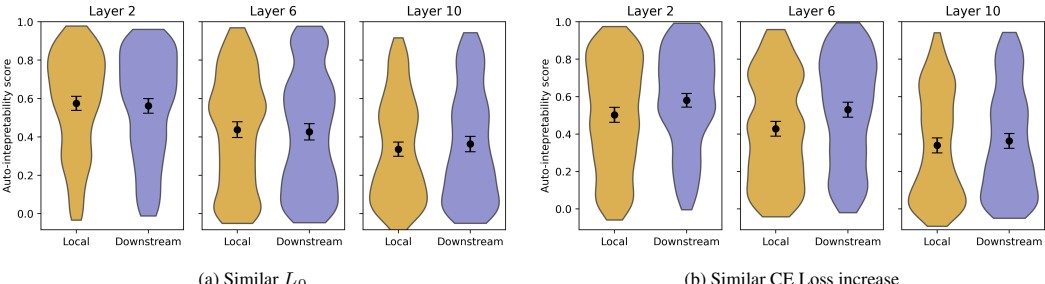

(a) Similar $L_0$
(b) Similar CE Loss increase

Figure 9: Comparison of auto-interpretability scores between $SAE_{e2e+ds}$ and $SAE_{local}$ for runs with similar $L_0$ (see Table 3) and similar CE loss increases (see Table 2). Error bars are a bootstraped 95% confidence interval for the true mean auto-interpretability scores. Measured on approximately $200(\pm2)$ randomly selected features per dictionary.

Table 4: Estimates of the difference between the mean auto-interpretability scores for $SAE_{e2e+ds}$ and $SAE_{local}$ (Figure 9). A positive difference indicates $SAE_{e2e+ds}$ is more interpretable. For each comparison we use bootstrapping to compute a 95% confidence interval and a two-tailed p-value that the means are equal.

|               | Layer | Mean diff. | 95% CI          | p-value |
|---------------|-------|------------|-----------------|---------|
| Similar $L_0$ | 2     | $-0.01$    | $[-0.07, 0.04]$ | 0.61    |
|               | 6     | $-0.01$    | $[-0.07, 0.05]$ | 0.71    |
|               | 10    | 0.03       | $[-0.03, 0.08]$ | 0.33    |
| Similar CE    | 2     | 0.08       | $[0.02, 0.13]$  | 0.0057  |
|               | 6     | 0.10       | $[0.05, 0.16]$  | 0.00044 |
|               | 10    | 0.02       | $[-0.03, 0.08]$ | 0.41    |

# B   Analysis of reconstructed activations

We saw in Appendix A.4 that our e2e-trained SAEs are much worse at reconstructing the exact activation compared to locally-trained SAEs. We performed some initial analysis of why this is.

## B.1   Scale

A common problem with SAEs is "feature-supression", where the SAE output has considerably smaller norm than the input [Wright and Sharkey, 2024, Rajamanoharan et al., 2024]. We observe this as well, as shown in Figure 10 for an SAE$_{e2e+ds}$ in layer 6. Note the cluster of activations with original norm around 3000; these are the activations at position 0.

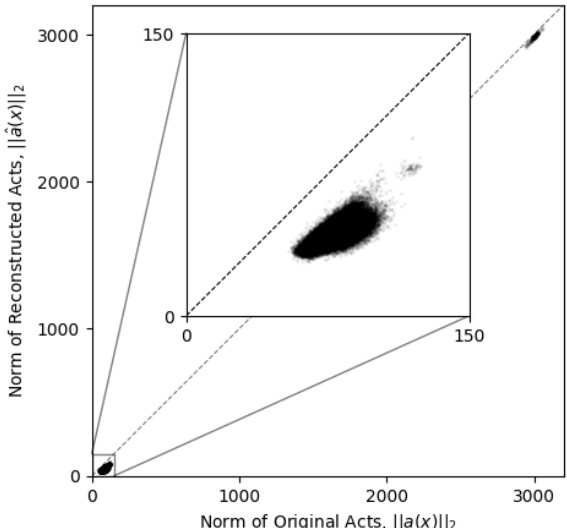

Figure 10: A scatterplot showing the $L_2$-norm of the input and output activations for out SAE$_{e2e+ds}$ in layer 6.

Table 5: $L_2$ Ratio for the SAEs of similar CE loss increase, as in Table 2.

| Layer | Position 0 | | | Position $> 0$ | | |
|---|---|---|---|---|---|---|
| | 2 | 6 | 10 | 2 | 6 | 10 |
| local | 1.00 | 1.00 | 1.00 | 0.98 | 0.92 | 0.91 |
| e2e | 0.16 | 0.11 | 0.08 | 0.67 | 0.31 | 0.15 |
| downstream | 0.14 | 0.99 | 0.99 | 0.74 | 0.56 | 0.32 |

We can measure suppression with the metric:

$$L_2 \text{ Ratio} = \mathbb{E}_{x \in \mathcal{D}} \frac{||\hat{a}(x)||}{||a(x)||}$$

which is presented in Table 5 for all of the similar CE loss increase SAEs in Table 2. Generally, SAE$_{e2e}$ has the most feature-suppression. This is as layer-norm is applied to the residual stream before the activations are used, which can allow the network to re-normalize the downscaled activations and keep similar outputs. The downscaled activations will still disrupt the normal ratio between the residual stream before the SAE is applied and the outputs of future layers that are added to the residual stream.

## B.2   Direction

Both SAE$_{e2e}$ and SAE$_{e2e+ds}$ do significantly worse at reconstructing the directions of the original activations than SAE$_{local}$ (Figure 11). Note, however, that we are comparing runs with similar CE

loss increases. $SAE_{local}$ is the only one of the three that is trained directly on reconstructing these activations, and achieves this reconstruction with significantly higher average $L_0$.

Overall, $SAE_{e2e+ds}$ and $SAE_{e2e}$ reconstruct the activation direction in the current layer similarly well, with $SAE_{e2e+ds}$ doing better at layer 6 and but worse at layer 10.

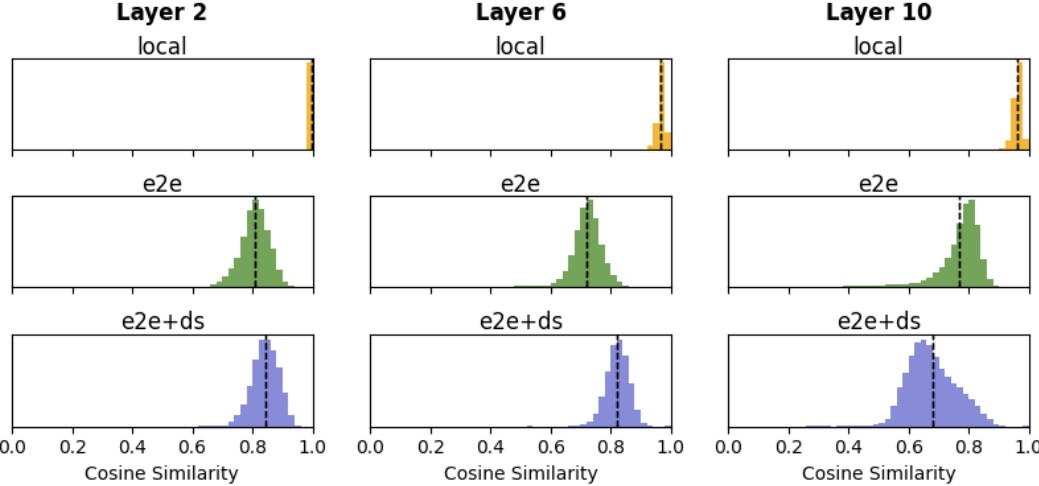

Figure 11: Distribution of cosine similarities between the original and reconstructed activations, for our SAEs with similar CE loss increases (Table 2). We measure 100 sequences of length 1024.

## B.3 Explained variance

How much of the reconstruction error seen earlier (Section 3.2) is due to feature shrinkage? One way to investigate this is to normalize the activations of the SAE output before comparing them to the original activation.[6] In Figure 12, we compare the explained variance for the reconstructed activations of each type of SAE in layer 6, both with and without normalizing the activations first. Normalizing the activations greatly improves the explained variance of our e2e SAEs. Despite this, the overall story and relative shapes of the curves are similar.

---

[6] To "normalize" we apply center along the embedding dimension and scale the resulting vector to have unit norm. This is equivalent to Layer Normalization with no affine transformation. We use this as Layer Normalization is applied to the residual-stream activations before they are used by the network.

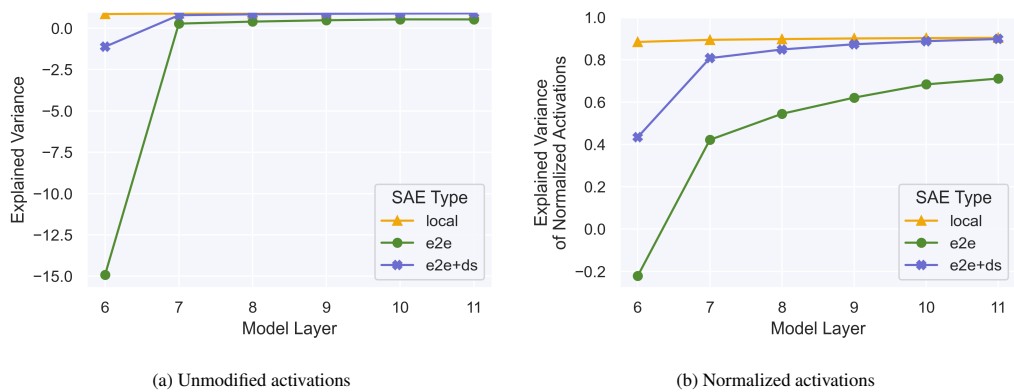

(a) Unmodified activations          (b) Normalized activations

Figure 12: Explained variance between activations from the model with and without the SAE inserted. Measured at all later layers for our set of SAEs with similar CE loss increase in layer 6 (Table 1). In (b) we apply Layer Normalization to the activations before comparison.

# C   Effect of gradient norms on the number of alive dictionary elements

One of our goals is to reduce the total number of features needed over a dataset (i.e. the alive dictionary elements), thereby reducing the computational overhead of any method that makes use of these features. We showed in Figure 4 that $SAE_{e2e}$ and $SAE_{e2e+ds}$ consistently use fewer dictionary elements for the same amount of performance when compared with $SAE_{local}$. We also see that $SAE_{e2e+ds}$ uses fewer elements than $SAE_{e2e}$ for layers 2 and 6 but not layer 10.

Notice in Table 2, however, that the number of alive dictionary elements is negatively correlated with the norm of the gradients during training. This begs the question: If we increase the learning rate, is it possible to maintain performance in $L_0$ vs CE loss increase while also decreasing the number of alive dictionary elements?

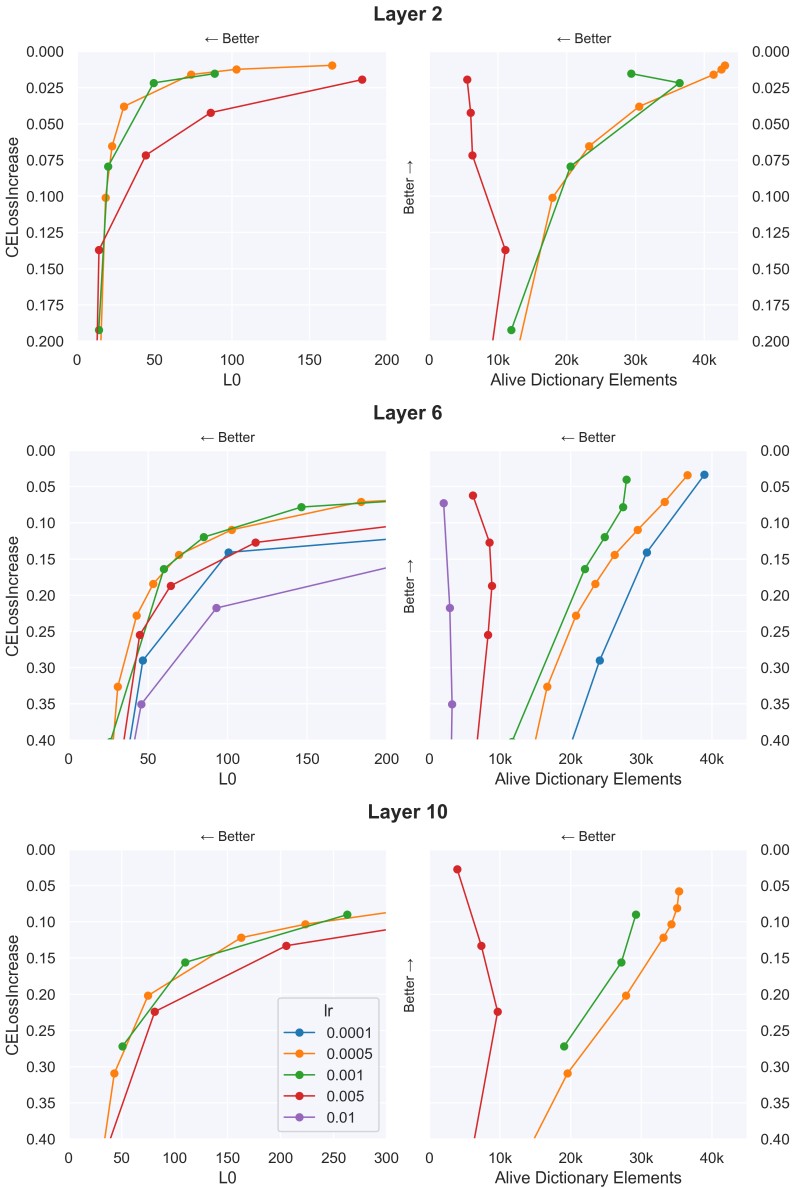

Figure 13: Varying the learning rate for $SAE_{local}$ on layers 2, 6 and 10. All other parameters are the same as the local runs listed in the similar CE loss increase Table 2.

In Figures 13, 14, 15, we show the effect that varying the learning rate has on performance for $SAE_{local}$, $SAE_{e2e}$, and $SAE_{e2e+ds}$, respectively. In all cases, we see that learning rates higher than our

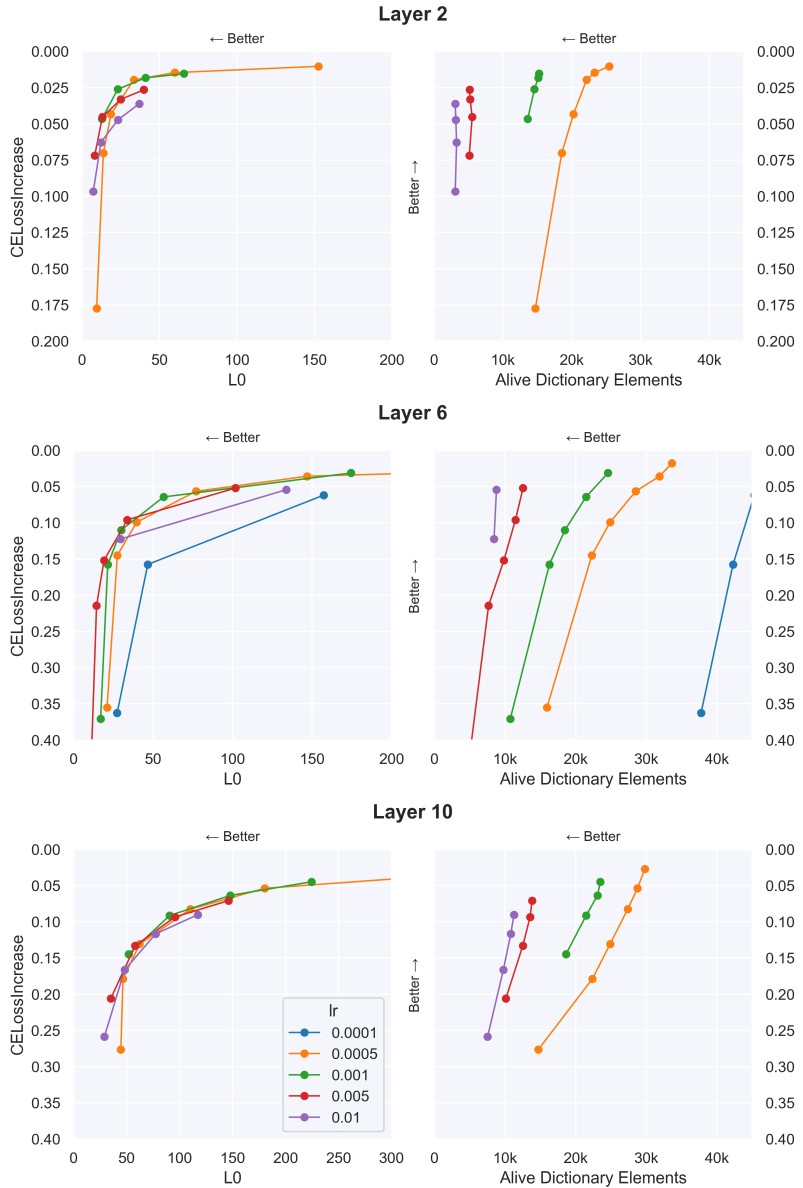

Figure 14: Varying the learning rate for $SAE_{e2e}$ on layers 2, 6 and 10. All other parameters are the same as the local runs listed in the similar CE loss increase Table 2.

default of $0.0005$ require fewer dictionary elements for the same level of performance on CE loss increase. We also see that these runs with higher learning rates (up to a limit) can have a better $L_0$ vs CE loss increase frontier at high sparsity levels and is similar or worse at low sparsity levels.

This effect appears to be more pronounced for $SAE_{e2e}$ and $SAE_{e2e+ds}$ than $SAE_{local}$, indicating that e2e SAEs may require even fewer alive dictionary elements compared to $SAE_{local}$s than what is presented in the figures in the main text.

While not shown in these figures, a downside of using learning rates larger than $0.0005$ is that it can cause the $L_0$ metric to steadily increase during training after an initial period of decreasing. This occurred for all of our SAE types, and was especially apparent in later layers. Due to this instability, we persisted with a learning rate of $0.0005$ for our main experiments. We expect that training tweaks such as using a sparsity schedule could help remedy this issue and allow for using higher learning rates.

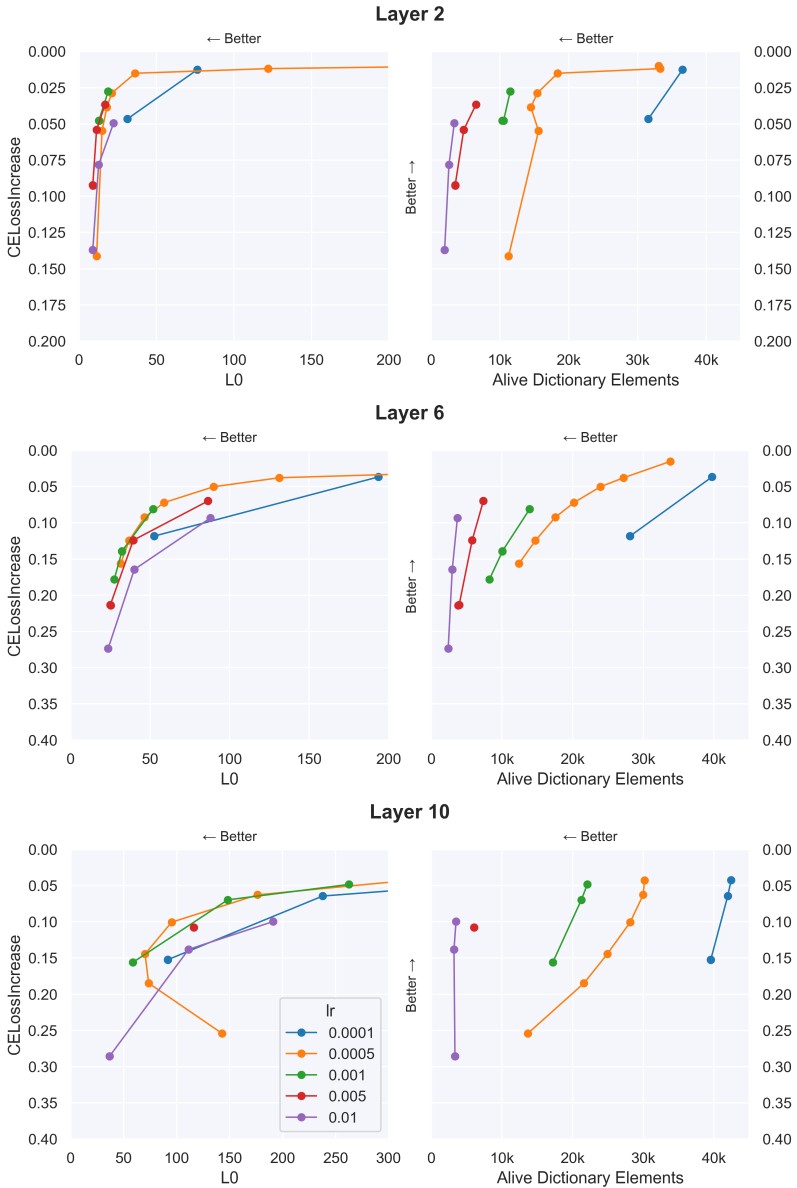

Figure 15: Varying the learning rate for SAE$_{\text{e2e+ds}}$ on layers 2, 6 and 10. All other parameters are the same as the local runs listed in the similar CE loss increase Table 2.

# D   Experimental details and hyperparameters

Our architectural and training design choices were selected with the goal of maximizing $L_0$ vs CE loss increase Pareto frontier of $\text{SAE}_{\text{local}}$. We then used the same design choices for $\text{SAE}_{\text{e2e}}$ and $\text{SAE}_{\text{e2e+ds}}$. Much of our design choice iteration took place on the smaller Tinystories-1m due to time and cost constraints.

Our SAE encoder and decoder both have a regular, trainable bias, and use Kaiming initialization. To form our dictionary elements, we transform our decoder to have unit norm on every forward pass. We do not employ any resampling techniques [Bricken et al., 2023] as it is unclear how these methods affect the types of features that are found, especially when aiming to find functional features with e2e training. We clip the gradients norms of our parameters to a fixed value (10 for GPT2-small). This only affects the very large grad norms at the start of training and the occasional spike later in training. We do not have strong evidence that this is worthwhile to do on GPT2-small, and it does comes at a computational cost.

We train for $400k$ samples of context size $1024$ on Open Web Text with an effective batch size of $16$. We use a learning rate of $5e-4$, with a warmup of $20k$ samples, a cosine schedule decaying to $10\%$ of the max learning rate, and the Adam optimizer [Kingma and Ba, 2017] with default hyperparameters.

For $\text{SAE}_{\text{e2e+ds}}$, we multiply our KL loss term by a value of $0.5$ in our implementation. Note that if we instead fixed this value to $1$ and varied the other loss coefficients, we would also need to vary other coefficients such as learning rate and effective batch size accordingly, which may have been difficult. This said, fixing this parameter to $1$ and having fewer overall hyperparemeters may be a better option going forward, as it turns out to be difficult to tune the other coefficients in this setting anyway. We set the *total_coeff* (i.e., the coefficient that multiplies the downstream reconstruction MSE, denoted $\beta$ in Equation 1) to $2.5$ for layers 2 and 6, and to $0.05$ for layer 10. Note from Equation 1 that this coefficient gets split evenly among all downstream layers. It's likely that a different weighting of these parameters is more desirable, but we did not explore this for this report.

It's worth noting that we did not iterate heavily on loss function design for $\text{SAE}_{\text{e2e+ds}}$, so it's likely that other configurations have better performance (e.g. having different downstream reconstruction loss coefficients depending on the layer, and/or including the reconstruction loss at the layer containing the SAE).

Note that in our loss formulation (Section 2), we divide our sparsity coefficient $\lambda$ by the size of the residual stream $\dim(a^{(l)}(x))$. This is done in an attempt to make our sparsity coefficient robust to changes in model size. The idea is that the $L_1$ score for an optimal SAE will be a function of the size of the residual stream. However, we did not explore this relationship in detail and expect that other functions of residual stream size (and perhaps dictionary size) are more suitable for scaling the sparsity coefficient.

For GPT2-small we stream the dataset https://huggingface.co/datasets/apollo-research/Skylion007-openwebtext-tokenizer-gpt2 which is a tokenized version of OpenWebText ([Gokaslan and Cohen, 2019]) (released under the license CC0-1.0). The tokenization process is the same as was used in GPT2 training, with a 'BOS' token between documents.

We evaluate our models on $500$ samples of the Open Web Text dataset (a different seed to that used for training). We consider a dictionary element *alive* if it activates at all on $500k$ training tokens.

Note that information from all of our runs are accessible in this Weights and Biases ([Biewald, 2020]) report, including the weights, configs and numerous metrics tracked throughout training. The SAEs from these runs can be loaded and further analysed in our library https://github.com/ApolloResearch/e2e_sae/.

We used NVIDIA A100 GPUs with $80$GB VRAM (although the GPU was saturated when using smaller batch sizes that used $40$GB VRAM or less).

Our library imports from the TransformerLens library [Nanda and Bloom, 2022] (released under the MIT License), which is used to download models via HuggingFace's Transformers library [Wolf et al., 2020] (released under the Apache License 2.0). GPT2-small is released under the MIT license. The Tinystories-1M model is released under the Apache License 2.0 and it's accompanying dataset is released under CDLA-Sharing-1.0.

# E   Varying initial dictionary size and number of training samples

## E.1   Varying initial dictionary size

In Figure 16 we show the effect of varying the initial dictionary size for our layer 6 similar CE loss increase SAEs in Table 2. For all SAE types, we see $L_0$ vs CE loss increase improve with diminishing returns as the dictionary size is scaled up, capping out at a dictionary size of roughly 60. This comes at the cost of having more alive dictionary elements with increasing dictionary size.

It's worth mentioning that, after preliminary investigation on Tinystories-1M, it's possible to reduce the dictionary ratio to 5 times the residual stream and still achieve a good $L_0$ vs CE loss increase tradeoff, as well as reducing the number of alive dictionary elements. See this Weights and Biases report for details https://wandb.ai/sparsify/tinystories-1m-ratio/reports/Scaling-dict-size-tinystories-blocks-4-layerwise--Vmlldzo3MzMzOTcw.

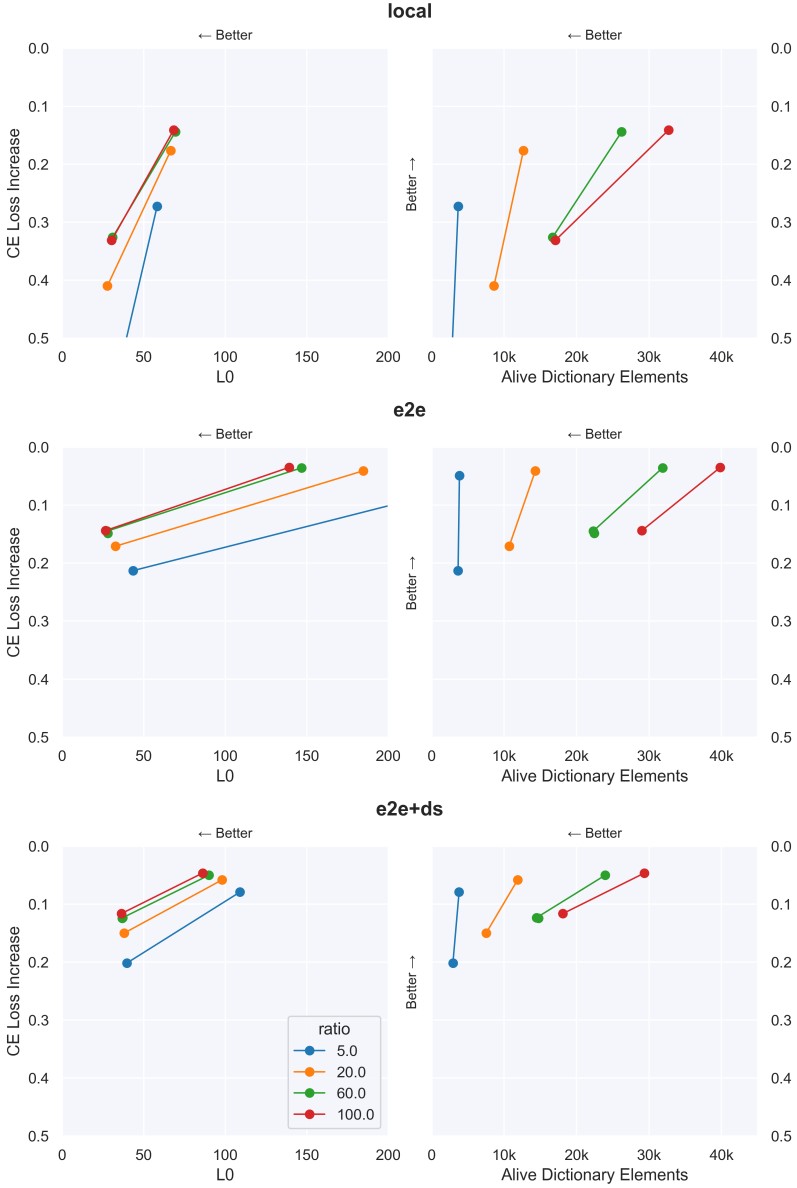

Figure 16: Sweep over the SAE dictionary size for layer 6 (where 'ratio' is the size of the initial dictionary divided by the residual stream size of 768). All other parameters are the same as in the similar CE loss increase runs in Table 2.

## E.2 Varying number of training samples

In Figure 17 we analyse the effect of varying the number of training samples for each SAE type on layer 6 of our similar CE loss increase SAEs. For $SAE_{local}$, training for 50k samples is clearly insufficient. The difference between training on 200k, 400k, and 800k samples is quite minimal for both $L_0$ vs CE loss increase and alive_dict_elements vs CE loss increase.

For $SAE_{e2e}$, we see improvements to $L_0$ vs CE loss increase when increasing from 50k to 800k samples but with diminishing returns. In contrast to $SAE_{local}$, we see a steady improvement in alive_dict_elements vs CE loss increase as we increase the number of samples. Note that training $SAE_{e2e}$ or $SAE_{e2e+ds}$ for 800k samples takes approximately 23 hours on a single A100.

For $SAE_{e2e+ds}$, the $L_0$ vs CE loss increase and alive_dict_elements vs CE loss increase improves up until 400k samples where performance maxes out.

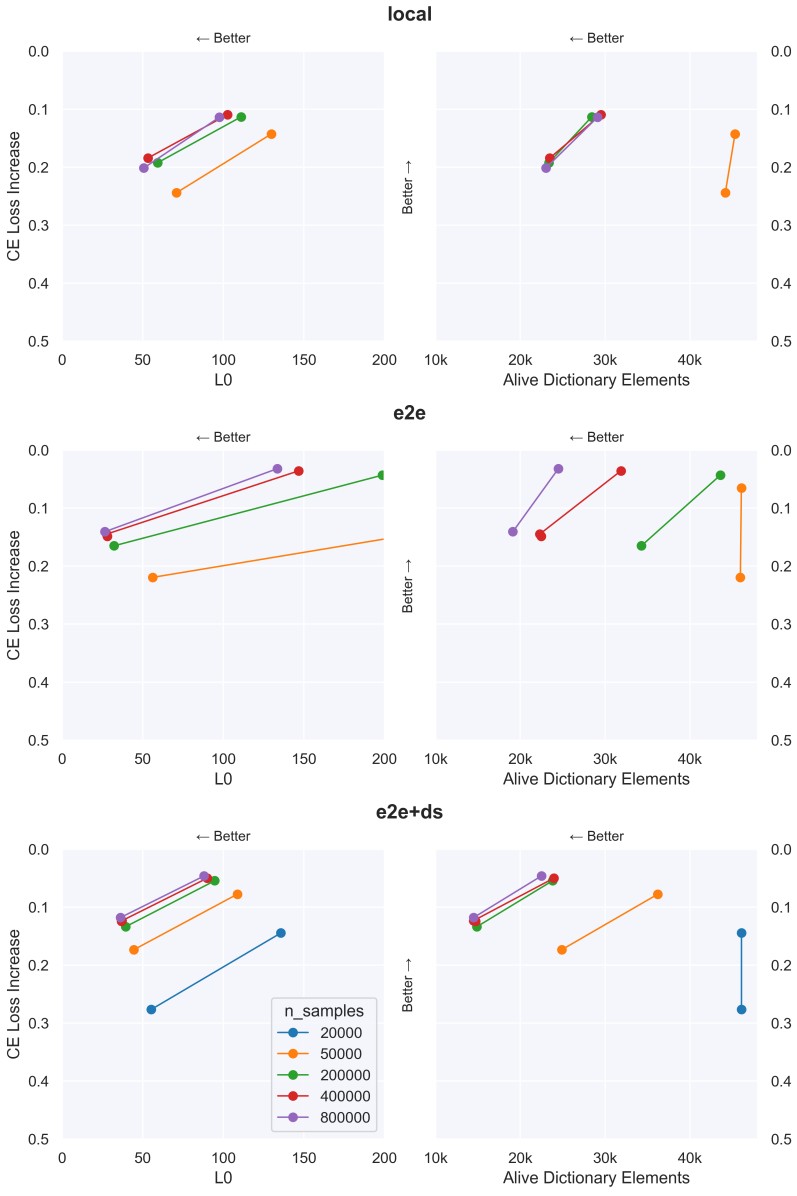

Figure 17: Sweep over number of samples trained on layer 6. All other parameters are the same as in the similar CE loss increase runs in Table 2.

# F  Robustness of features to different seeds

We show in Figure 18 that, for a variety of sparsity coefficients and layers, our training runs are robust to the random seed. Note that the seed is responsible for both SAE weight initialization as well as the dataset samples used in training and evaluation.

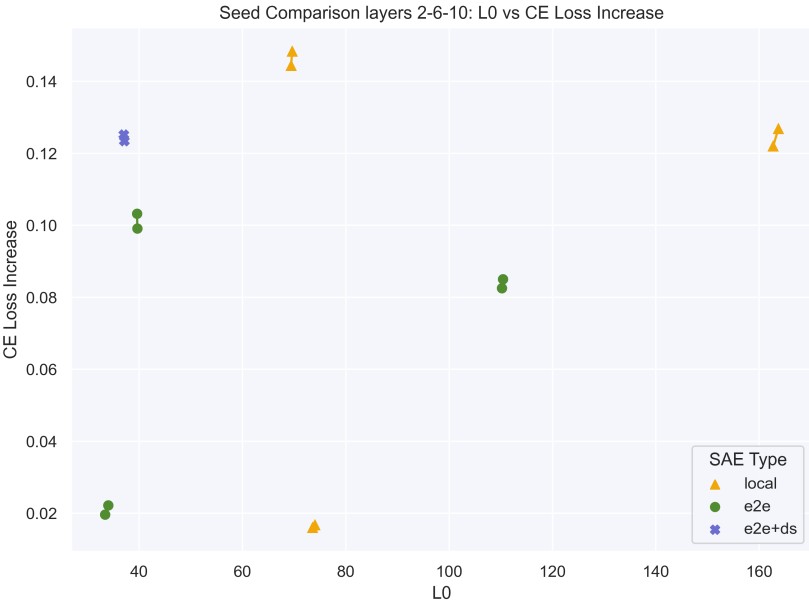

Figure 18: A sample of SAEs for layers 2, 6 and 10 for all run types showing the robustness of SAE training to two different seeds.

# G    Analysis of UMAP plots

To explore the qualitative differences between the features learned by SAE$_{\text{local}}$ and SAE$_{\text{e2e+ds}}$, we first visualize the SAE features using UMAP [McInnes et al., 2018] (Figures 19, 20).

## G.1    UMAP of layer $6$ SAEs

Although there is substantial overlap between the features from both types of SAE in the plot, there are some distinct regions that are dense with SAE$_{\text{e2e+ds}}$ features but void of SAE$_{\text{local}}$, and vice versa. We look at the features in these regions along with features in other identified regions of interest such as small mixed clusters in layer 6 of GPT2-small in more detail. We label the regions of interest from A to G in Figure 19, and provide human-generated overview of these features below. Features from this UMAP plot can be explored interactively at `https://www.neuronpedia.org/gpt2sm-apollojt`. For each region, we also share links to lists of features in that region which go to an interactive dashboards on Neuronpedia.

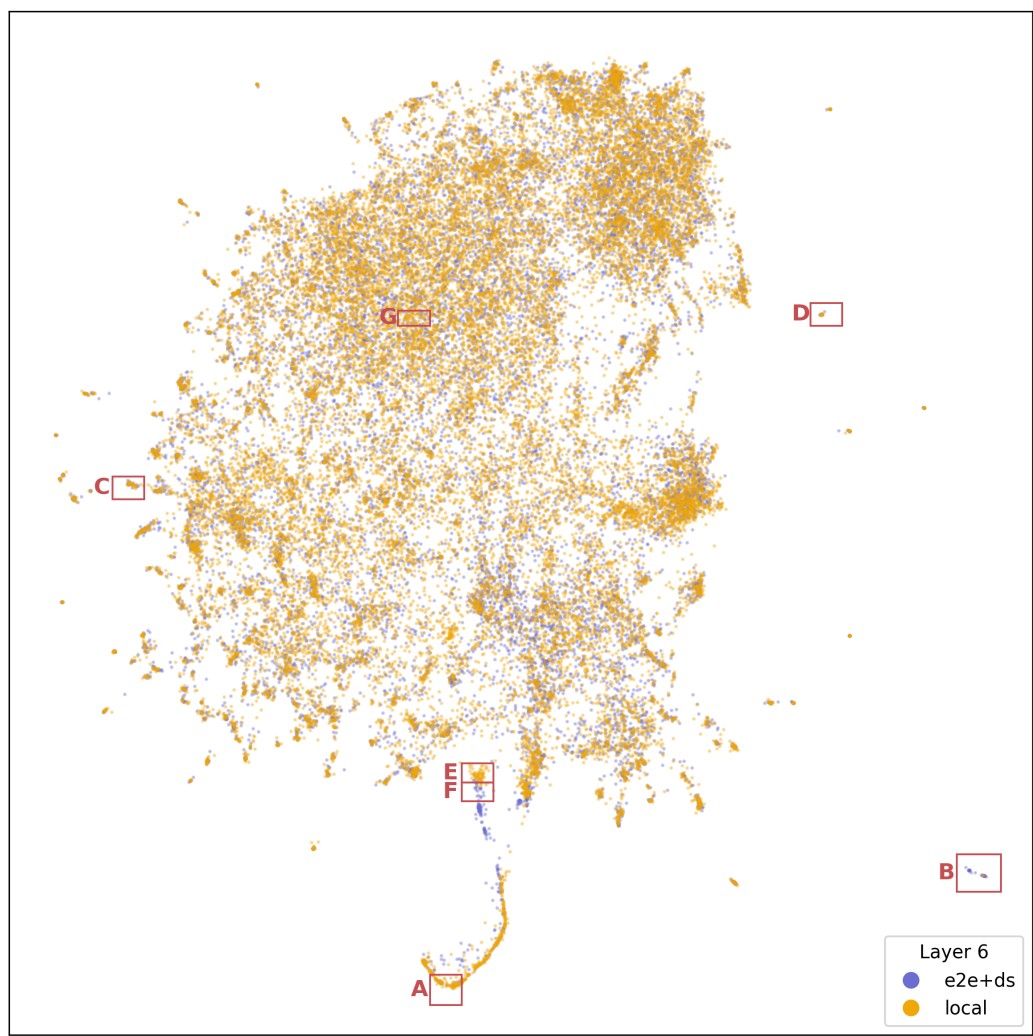

Figure 19: UMAP plot of SAE$_{\text{e2e+ds}}$ and SAE$_{\text{local}}$ features for layer 6 on runs with similar CE loss increase in GPT2-small.

**Region A (SAE$_{e2e+ds}$ features (18). SAE$_{local}$ features (91))**

Many of these features appear to be late-context positional features, or miscellaneous tokens that only activate in particularly late context positions. It may be the case that SAE$_{e2e+ds}$ has fewer positional features than local, as indicated by the 18 SAE$_{e2e+ds}$ vs 91 SAE$_{local}$ local features in this region (and similar in surrounding reasons). This said, we have not ruled out whether positional features for SAE$_{e2e+ds}$ are found elsewhere in the UMAP plot.

**Region B (SAE$_{e2e+ds}$ features (48). SAE$_{local}$ features (2))**

This region mostly contains features which activate on `<|endoftext|>` tokens, in addition to some newline and double newline. These are tokens that mark the beginning of a new context. Seemingly SAE$_{e2e+ds}$ contains many more distinct features for `<|endoftext|>` than SAE$_{local}$.

**Region C (SAE$_{e2e+ds}$ features (20). SAE$_{local}$ features (31))**

Region C potentially suggests more feature splitting happening in SAE$_{local}$ than SAE$_{e2e+ds}$. For SAE$_{e2e+ds}$, each feature activates most strongly on tokens "by" or "from" in a broad range of contexts. For SAE$_{local}$, each feature activates most strongly in fine-grained contexts, such as "goes by" vs "led by" vs ". By" vs "stop by" vs "<media>, by author" vs "despised by" vs "overtaken by" vs "issued by" vs "step-by-step / case-by-case / frame-by-frame" vs "Posted by" vs "Directed by" vs "killed by" vs "by".

**Region D (SAE$_{e2e+ds}$ features (11). SAE$_{local}$ features (19))**

These features all activate on "at" in various contexts. As in Region C, the SAE$_{e2e+ds}$ features appear less fine-grained. Examples of SAE$_{local}$ features not present in SAE$_{e2e+ds}$: "Announced at" or "presented at" or "revealed at" feature (https://www.neuronpedia.org/gpt2-small/6-res_scl-ajt/40197). "At" in technical contexts (https://www.neuronpedia.org/gpt2-small/6-res_scl-ajt/34541).

**Region E (SAE$_{e2e+ds}$ features (3). SAE$_{local}$ features (67))**

All features appear to boost starting words which would come after a paragraph or a full stop to start a new idea, such as "Finally", "Moreover", "Similarly", "Furthermore", "Regardless", "However" and so on. They seem to be differentiated by perhaps activating in different contexts. For example https://www.neuronpedia.org/gpt2-small/6-res_scl-ajt/4284 activates on full stops and newlines in technical contexts so it can predict things like "Additionally", "However", and "Specifically". On the other hand, https://www.neuronpedia.org/gpt2-small/6-res_scl-ajt/13519 activates on full stops in baking recipes so it can predict things like "Then", "Afterwards", "Alternatively", "Depending" and so on.

**Region F (SAE$_{e2e+ds}$ features (19). SAE$_{local}$ features (8))**

These seem mostly similar to Region E. It's not clear what distinguishes the regions looking at the feature dashboards alone.

**Region G (SAE$_{e2e+ds}$ features (41). SAE$_{local}$ features (71))**

The features in both SAEs seem to activate on fairly specific different words or phrases. There is no obvious distinguishing features. It's possible that SAE$_{e2e+ds}$ features tend to activate more specifically and on fewer tokens than the corresponding SAE$_{local}$ features. An example of this can be seen when comparing https://www.neuronpedia.org/gpt2-small/6-res_scefr-ajt/13910 (a SAE$_{e2e+ds}$ feature), with https://www.neuronpedia.org/gpt2-small/6-res_scl-ajt/45568 (a SAE$_{local}$ feature).

### G.2 UMAP of layer 2 and layer 10 SAEs

In Figure 20, we show UMAP plots for layers 2 and 10. We interpret a single region from layer 10 in the next section.

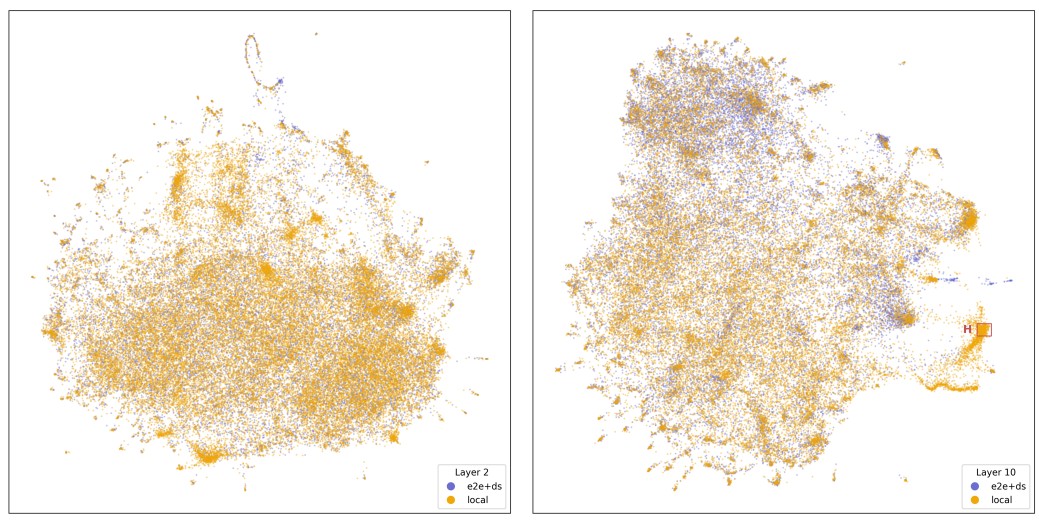

Figure 20: UMAP of $SAE_{e2e+ds}$ and $SAE_{local}$ features for layers 2 and 10 on runs with similar CE loss increase in GPT2-small.

### G.3 Region H in layer 10 ($SAE_{e2e+ds}$ features (2). $SAE_{local}$ features (593))

While some individual features in this region are interpretable, there is no obvious uniting theme semantically. There is, however, a *geometric* connection. In particular, these are features that point away from the 0th PCA direction in the original model's activations (Figure 21).

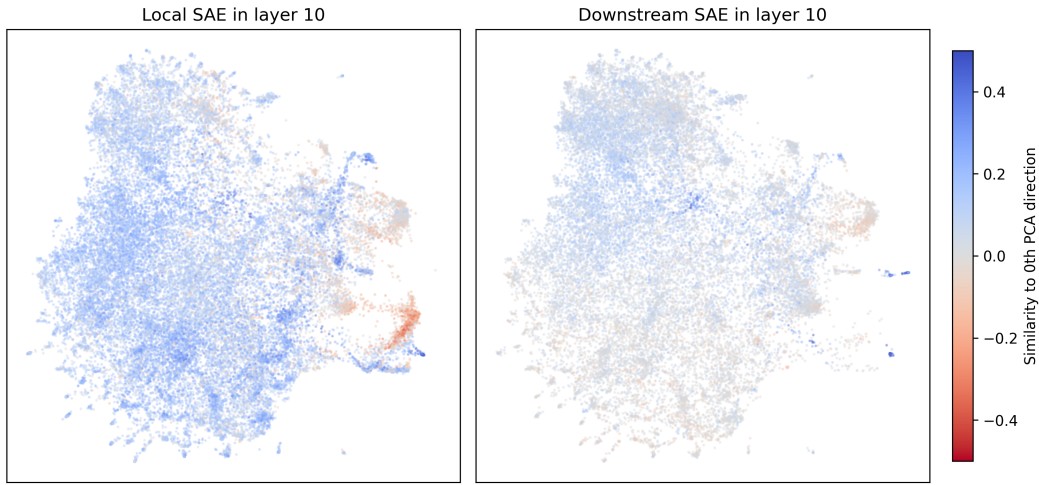

Figure 21: The UMAP plot for $SAE_{e2e+ds}$ and $SAE_{local}$ directions, with points colored by their cosine similarity to the 0th PCA direction.

The 0th PCA direction is nearly exactly the direction of the outlier activations at position 0 (see also Appendix B). Activations in this direction are tri-modal, with large outliers at position 0 and smaller outliers at end-of-text tokens (Figure 22).

We can measure how well an SAE preserves a particular direction by measuring the correlation between the input and output components in that direction. Our $SAE_{e2e+ds}$ faithfully reconstructs the activations in this direction at position 0 ($r = 0.996$), but not at other positions ($r = 0.262$). This is a particularly poor reconstruction compared to $SAE_{local}$ or other PCA directions (Figure 23a).

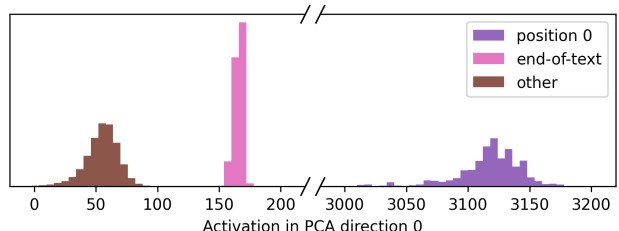

Figure 22: A histogram of the 0th PCA component of the activations before layer 10.

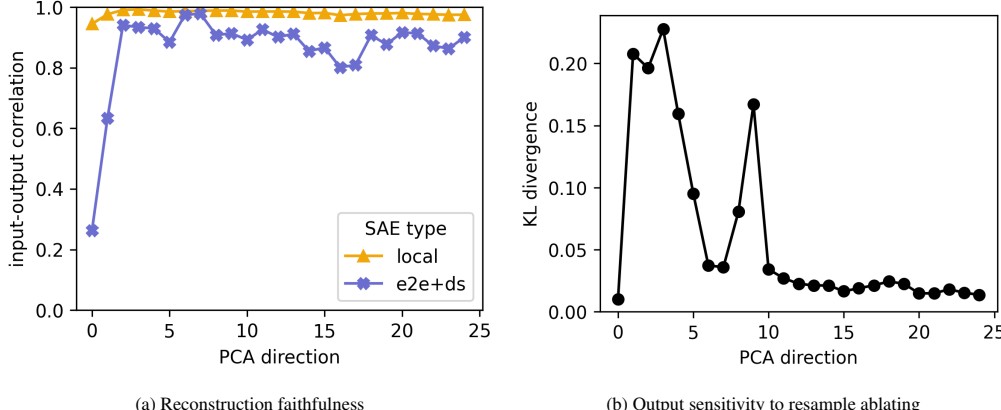

(a) Reconstruction faithfulness

(b) Output sensitivity to resample ablating

Figure 23: For each PCA direction before layer 10 we measure two qualities. The first is how faithfully $\text{SAE}_{\text{local}}$ and $\text{SAE}_{\text{e2e+ds}}$ reconstruct that direction by measuring correlation coefficient. The second is how functionally-important the direction is, as measured by how much the output of the model changes when resample ablating the direction.

$\text{SAE}_{\text{e2e+ds}}$'s poor reconstruction of the activations in this direction implies that the differences may not be functionally relevant. We can measure this by *resample ablating* the activation in this direction at all non-zero positions. This means we perform the following intervention in a forward hook:

$$a(x)_i \leftarrow a(x)_i - Pa(x)_i + Pa(x')_j$$

Where $a(x)_i$ is the activation at position $i > 0$, $a(x')_j$ is the resampled activation for a different input $x'$ and position $j > 0$, and $P$ is a projection matrix onto the 0th PCA direction.

After performing this ablation, the kl-divergence from the original activations is only 0.01. This difference is smaller than repeating the experiment for any other direction in the first 30 PCA directions (Figure 23b).

This means that the exact value of this component of the activation (at positions $> 0$) is mostly functionally irrelevant for the model. $\text{SAE}_{\text{local}}$ still captures the direction faithfully, as it is purely trained to minimize MSE. While $\text{SAE}_{\text{e2e+ds}}$ fails to preserve this direction accurately, this seems to allow it to have a cleaner dictionary, avoiding $\text{SAE}_{\text{local}}$'s cluster of features that point partially away from this direction.

# H  Training time

The training time for each type of SAE in GPT2-small is shown in Table 6. We see that e2e SAEs are 2-3.5x slower than SAE$_{local}$. Note that one can reduce training time with little performance cost by training on fewer that 400k samples (Figure 17) and/or using an initial dictionary ratio of less than 60x the residual stream size (Figure 16). Using locally trained SAEs as initialization for e2e SAEs or training multiple SAEs at different layers concurrently are also possible solutions.

Table 6: Training times for different layers and SAE training methods using a single NVIDIA A100 GPU on the residual stream of GPT2-small at layer 6. All SAEs are trained on 400k samples of context length 1024, with a dictionary size of 60x the residual stream size of 768.

| Layer | SAE$_{local}$ | SAE$_{e2e}$ | SAE$_{e2e+ds}$ |
|-------|--------------|-------------|----------------|
| 2     | 3h 45m       | 12h 24m     | 12h 30m        |
| 6     | 4h 45m       | 11h 20m     | 11h 24m        |
| 10    | 5h 19m       | 10h 12m     | 10h 20m        |

Table 7: Faithfulness on subject-verb agreement when replacing the activations with SAE outputs.

| | | (a) Similar CE | | | | | | (b) Similar $L_0$ | | | |
| Layer | Type | Simple | Across PP | Across RC | Within RC | Layer | Type | Simple | Across PP | Across RC | Within RC |
|---|---|---|---|---|---|---|---|---|---|---|---|
| 2 | local | 101.0% | 103.1% | 102.4% | 100.6% | 2 | local | 103.4% | 107.7% | 105.7% | 101.0% |
| | e2e | 104.1% | 100.5% | 98.8% | 102.1% | | e2e | 100.8% | 101.0% | 101.6% | 98.9% |
| | e2e+ds | 105.6% | 101.9% | 99.1% | 102.2% | | e2e+ds | 104.8% | 109.3% | 97.9% | 98.8% |
| 6 | local | 100.6% | 96.0% | 79.0% | 102.8% | 6 | local | 99.6% | 93.2% | 89.1% | 99.1% |
| | e2e | 96.8% | 94.5% | 104.7% | 94.4% | | e2e | 101.2% | 101.3% | 107.0% | 101.1% |
| | e2e+ds | 95.2% | 79.2% | 95.5% | 98.7% | | e2e+ds | 95.2% | 79.2% | 95.5% | 98.7% |
| 10 | local | 97.2% | 79.3% | 85.2% | 106.0% | 10 | local | 95.1% | 77.8% | 68.6% | 106.9% |
| | e2e | 92.5% | 90.5% | 89.1% | 100.0% | | e2e | 92.5% | 90.5% | 89.1% | 100.0% |
| | e2e+ds | 104.7% | 107.4% | 84.2% | 107.4% | | e2e+ds | 104.7% | 107.4% | 84.2% | 107.4% |

# I Faithfulness of SAEs on subject verb agreement task

Our main evaluation metrics presented in Section 3 measure the functional importance of the features learned by the SAEs on the next-token language modeling task used to train the model. We also experimented with evaluating the SAEs on a downstream task: how faithfully the dictionaries represent the information the model uses to perform subject-verb agreement. This task is directly inspired by the analysis in Marks et al. [2024].

## I.1 Methodology

We use datasets from Finlayson et al. [2021] with 4 variations of a subject-verb agreement task:

- Simple: *The parent/s is/are*
- Across participle phrase (PP): *The secretary/secretaries near the cars has/have*
- Within relative clause (RC): *The athlete that the manager/managers likes/like*
- Across RC: *The athlete/athletes that the managers like do/does*

For each template, we use 1000 datapoints with different subjects and verbs. For each input, we can compute the logit difference that the model assigns to the correct and incorrect forms of the verb.

Following Marks et al. [2024] we compute the *faithfulness* of this logit difference when intervening on the network's activations. Let $m$ represent the mean logit difference between the correct and incorrect verb forms across the dataset. Let $M$ be the original model and $\tilde{M}$ be the model under some intervention. We measure the faithfulness of the intervention as $\frac{m(\tilde{M})-m(\varnothing)}{m(M)-m(\varnothing)}$ where $\varnothing$ represents ablating the entire residual stream.

A faithfulness of $0\%$ thus means $\tilde{M}$ performs no better than random, while a faithfulness of $100\%$ means the intervention does not change performance. Faithfulness numbers greater than $100\%$ mean the model is, on average, more confident in the correct verb with the intervention than without.

## I.2 Faithfulness with complete SAEs

We first test the faithfulness of the models with the SAEs inserted (Table 7). An SAE which preserves functionally relevant features in the activations would have faithfulness close to $100\%$ on all tasks the model is trained to perform.

While all SAEs preserve most of the logit-difference, there is significant variation across SAE types, layers, and tasks. The local SAEs in layer 10 have the worst faithfulness, although SAE_{e2e+ds} in layer 6 also has poor faithfulness across participial phrases.

## I.3 Faithfulness with a small number of features

We are also interested in if end-to-end training helps concentrate the functional relevance into a few specific SAE features. We thus ranked SAE features in Layer 6 Similar $L_0$ SAEs by their indirect

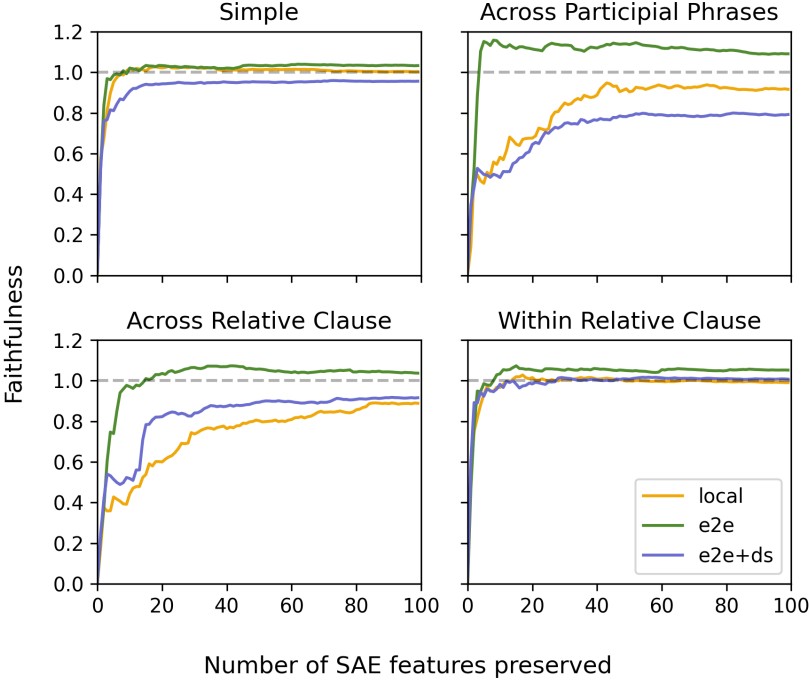

Figure 24: Faithfulness when mean-ablating all SAE features except the top $k$ with largest indirect effect. Experiments done with the similar $L_0$ SAEs on layer 6.

effect, measured by the change in faithfulness score when mean ablating that feature.[7] We then mean ablate all SAE features except the $k$ with largest indirect effect (Figure 24). We see $SAE_{e2e}$ needs comparatively few features to achieve high faithfulness scores, but $SAE_{local}$ and $SAE_{e2e+ds}$ have roughly similar curves.

## I.4  Discussion

Despite certain (SAE type, layer) combinations showing superior performance, there are no clear patterns between SAE types across tasks. These results indicate that e2e SAEs do not provide an obvious benefit on the selected downstream tasks. While e2e SAEs demonstrate benefits for the language modeling task on the full OpenWebText distribution, further work would be needed to find specific tasks and sub-distributions in which they provide the most benefit. Or, perhaps these sort of task-specific results are very noisy and it would be necessary to aggregate across many tasks and templates to differentiate between SAE training methodologies.

---

[7]When mean ablating an SAE feature we take it's mean *position-specific* value.

