# OpenReview forum: "Identifying Functionally Important Features with End-to-End Sparse Dictionary Learning"
_NeurIPS.cc/2024/Conference — NeurIPS 2024 poster_

### Official Review · Reviewer_ErCK · 2024-06-12

**Soundness:** 4
**Presentation:** 4
**Contribution:** 3
**Rating:** 6
**Confidence:** 4

**Summary:**

This work describes two new type of sparse autoencoders (SAEs) that are trained with two new loss functions, L_{e2e} and L_{e2e + ds}. L_{e2e} loss penalizes the KL divergence of the original model vs. when the SAE is inserted, as opposed to existing SAE training techniques which instead optimize MSE with the original activations. L_{e2e + downstream} additionally seeks to minimize the MSE of downstream layers after the SAE is inserted vs. the original model. Both types of new SAE have a better L0 vs. CE tradeoff than naive SAEs, and e2e + downstream additionally has similar downstream reconstruction loss. Interpretability is no worse for the new SAEs. Finally, e2e+ds has similar geometric properties to naive SAEs, including discovering similar features.

**Strengths:**

- The new SAE techniques that the authors introduce are a Pareto improvement over existing techniques on L0 vs. CE loss. Furthermore, the techniques are exciting because they is in a different direction from recent improvements in training SAEs: it is possible that the new techniques can be combined with recent techniques like gated SAEs and the top-k activation function.
- The investigation of the geometry of the different SAE features is extremely interesting. Most notably, e2e + ds finds similar features to local and is reproducible over different seeds, and e2e has potentially less feature splitting.
- The new techniques result in features of similar interpretability as local, as determined by standard automated interpretably tests.

**Weaknesses:**

- The main evaluation metrics that show improvement (CE loss for both new SAEs and later layer reconstruction error for e2e + downstream) are the same ones that are being optimized. This is noted by the authors when they mention Goodhart's law, but it would greatly improve the paper if there were additional metrics for downstream SAE quality that were tested (e.g. some of the methods described in section 4.4).
- Feature shrinkage (shown in Table 5) seems much worse with the new SAE training methods. Moreover, even with e2e + ds SAEs, later layer reconstructions in Figure 12 are much worse than local. This seems potentially a problem for e.g. finding circuits or understanding interactions between multiple SAEs via feature similarity. This is another reason I think additional downstream metrics would be useful.

**Questions:**

- Building off the above weaknesses, could the authors add a measure of downstream SAE functional importance that is not CE loss?
- Why doesn't the original SAE learn the "left block" of features that SAE e2e + ds does? It would be great to see if e.g. these have a small or negative impact on the variance explained.
- Did the authors try training an SAE with KL divergence + the local reconstruction error (i.e. MSE of layer X reconstruction for an SAE trained on layer X)? It is a little concerning to me that the layer X variance explained is so strongly negative because it means the SAE's reconstruction is very far off distribution.
- Can you quantify how much farther the <e2e ds to local> difference is vs. the inter seed distance? It is hard to know how to interpret Appendix G without this context, as inter-seed SAEs have different features learned too.
- The ~0.15 CE loss e2e+downstream SAE looks closer to the other chosen SAEs than the 0.125 CE loss one, does it change things to run with the ~0.15 one?

Small points:
- Line 41 states "a significant amount of [causal effects of some features is] mediated by the reconstruction residual errors", but I believe the cited work shows that this is true for circuits as a whole, not features individually.
- Lines 133-136 are confusing. Why is it okay to "clear out" features from layer X, but not layers X + 1...? In general, a more clear discussion of the tradeoff between learning functionally important features and maintaining model behavior would help I think.

I would be happy to raise my score if some of these questions/concerns were addressed.

**Limitations:**

I believe the authors should more explicitly discuss the Goodhart's law problem I've discussed above and ideally the increased feature suppression as well. Otherwise, I think the limitations and impacts are well addressed.

---

> ### Author Rebuttal · Authors · 2024-08-06
>
> Thank you for your evidently detailed read of our paper and your thoughtful comments.
>
> ## Comments on Weaknesses
> Regarding evaluation metrics, we've taken your criticism on board and run a set of evaluations on a set of downstream tasks. These are documented in the Author Rebuttal at the top (results in the attached pdf).
>
> As for your observation about feature shrinkage appearing worse for e2e SAEs. We agree. It turns out you can shrink activations in the residual stream without changing the output much and e2e SAEs take advantage of this. Note that this is partially because of layernorm (as shown in Figure 12). Fortunately, one can address the feature shrinkage issue with some of the SAE variants that were released following our submission (and that you mention in the summary of your review), e.g. topk SAEs, Gated SAEs, JumpReLU. We agree with your summary that these improvements should be orthogonal to the improvements we find with e2e SAEs.
>
> ## Comments on Questions
> > Why doesn't the original SAE learn the "left block" of features that SAE e2e + ds does? It would be great to see if e.g. these have a small or negative impact on the variance explained.
>
> If you’re referring to what appears to be the left mode of the somewhat bimodal distribution in Figure 8, we could not find any meaningful similarities between these features unfortunately. Once the anonymity is lifted, we will be able to publish an interactive dashboard with these features.
>
> > Did the authors try training an SAE with KL divergence + the local reconstruction error (i.e. MSE of layer X reconstruction for an SAE trained on layer X)? It is a little concerning to me that the layer X variance explained is so strongly negative because it means the SAE's reconstruction is very far off distribution.
>
> Please see the comment about variations of e2e+ds in the Author Rebuttal at the top.
>
> > Can you quantify how much farther the <e2e ds to local> difference is vs. the inter seed distance? It is hard to know how to interpret Appendix G without this context, as inter-seed SAEs have different features learned too.
>
> Good point. As for the means of those distributions, the <e2e ds to local> cross-type similarity (Figure 3c bottom) has a mean of 0.73, and the <e2e ds> cross-seed similarity (Figure 3b bottom) has a mean of 0.78. Visually, we do see the mode to the left of the cross-type plots that doesn’t exist in cross-seed plots, indicating that there is a set of features which are learned by e2e+ds but not local (as mentioned in response to your earlier question, we weren’t able to interpret any common properties of this set of features). We’ve plotted a UMAP comparison for different seeds on layer 6. Unlike our cross-type UMAP plots in the paper (Figure 19), we don’t see clear regions where one seed is represented but the other isn’t (though, of course, reading deeply into UMAP plots is a source of potential confusion). We’ve attached the layer 6 cross-seed UMAP to the pdf in Figure 3 of the Author Rebuttal.
>
> > The ~0.15 CE loss e2e+downstream SAE looks closer to the other chosen SAEs than the 0.125 CE loss one, does it change things to run with the ~0.15 one?
>
> Keen eye. The reason we didn’t select this run for our “similar CE loss” set was simply because it was run after our other analyses (including the time-consuming interpretations in Appendix G and dashboard creation) merely to fill out the pareto frontier for a nicer plot. After your comment, we reran the analysis (not the interpretations) with this run and did not find any significant differences. We posted a sample comparing the cross-type similarity with the different runs in Figure 2 of the uploaded pdf in the Author Rebuttal at the top.
>
> > Line 41 states "a significant amount of [causal effects of some features is] mediated by the reconstruction residual errors", but I believe the cited work shows that this is true for circuits as a whole, not features individually.
>
> We agree that it could technically be possible that the circuit does have this property but that individual features in the circuit do not. Though we think it’s reasonable to extrapolate that the same is true for individual features. Nonetheless, we’ve updated the text to be more accurate, thanks for pointing this out.
>
> > Lines 133-136 are confusing. Why is it okay to "clear out" features from layer X, but not layers X + 1...? In general, a more clear discussion of the tradeoff between learning functionally important features and maintaining model behavior would help I think.
>
> Our initial concern was that adding a reconstruction loss term at the current layer would have a much larger impact on reconstructing non-functional features than if the loss term was added at downstream layers. Though we have come to agree that this difference is merely quantitative rather than qualitative. As mentioned in the "Other variations of e2e+ds" section in the Author Rebuttal, we unfortunately have not run the extensive sweeps on the e2e+ds variations needed to give an insight to this question empirically.
>
> ## Summary
> Thanks again for your valuable comments. Hopefully we've addressed your concerns/questions, and are keen to hear if there are cases where we have not.

---

> > ### Comment · Reviewer_ErCK · 2024-08-10
> >
> > Thank you for responding to my comments! Most of my concerns are addressed, but I remain concerned that the only real metric that shows improvement is the one that is being optimized (the new experiments are a pretty limited evaluation and don't show a compelling improvement). Some ideas might be using board games like in the recent work from [Karvonen et. al.](https://arxiv.org/abs/2408.00113), showing more faithful circuits over a wide dataset like the ones studied in [Marks et. al.](https://arxiv.org/abs/2403.19647), or something else. I believe without more compelling downstream metrics, I cannot raise my score, so I will keep it as is. Thank you again!

---

### Official Review · Reviewer_Hd7R · 2024-06-14

**Soundness:** 3
**Presentation:** 3
**Contribution:** 2
**Rating:** 5
**Confidence:** 2

**Summary:**

This paper proposes to train Sparse Autoencoders (SAEs) with an additional loss function term: The KL divergence between the original output distribution and the output distribution obtained when using model with the inserted SAEs. This additional loss pushes the SAEs to focus on functionally important features
The authors find that this setup requires fewer total features and fewer simultaneously active features per data point.

**Strengths:**

- the authors point out a weakness of previous SEAs (feature splitting) and propose an intuitive solution: learning functionally important features by minimizing the KL divergence between original output and output with inserted SAEs

- writing is mostly clear and easy to follow

- the authors discuss limitations of their approach like robustness to random seeds, reconstruction loss, training time

**Weaknesses:**

- I think the major weakness is that the contribution is not very groundbreaking.
- There are no new findings wrt interpretability, and the e2e SAEs require more compute to be trained
- while the authors discuss several findings (robustness to random seeds, reconstruction loss, training time) the reader is left waiting a bit for a conclusion or take home message
- the comparison to local SAEs is sometimes a bit unclear:
	- the authors state that "locally trained SAEs are capturing information about dataset structure that is not maximally useful for explaining the algorithm implemented by the network." but do not show that their e2e SAEs are significantly more interpretable or helpful at explaining the algorithm implemented by the network, instead the e2e SAEs are merely "at least as interpretable as SAElocal"
	- the features found by local SAEs are not the functionally important ones , but SAElocal and SAEe2e+ds features are somewhat similar so local SAEs can be used as initializations (line 233)?


- notation could be improved
	- N_dict_elements and d_hidden seem to belong more into pseudo code so it looks a bit odd to mix it with more traditional mathy notation
	- $\lambda$ is hidden in the $\phi$ parameter but then you show values for lambda in Table 1 and looking back at i.e. Eq. (1) it's not immediately clear what $\lambda$ is and why $\phi$ is not mentioned. I would suggest either putting $\lambda$ directly in the equations or citing values for $\phi$

**Questions:**

line 50: but we no longer optimize activation reconstruction. -> but you do use activation reconstruction for $SAE_{e2e+ds}$, don't you?

Maybe I just missed i but where in the network do you plug in the SAEs? the residual stream/decoder layer output?

You say that taking different computational pathways through subsequent layers of the network might be a problem, but do you have any evidence for this happening? is this just section 3.3.2?

While reading, I was wondering if taking different computational pathways through subsequent layers of the network is still a problem if only one SAE is trained at a time. There should be little incentive for computing additional information since the original network post SAE insertion would not have capacity to handle these anyway, so it would only matter for the output directly.


small things:
- line 68: not sure if it's necessary to mention goodhart's law since the target is not cross-entropy loss difference but interpretability so I would be more worried about this if you start optimizing interpretability measures
- line 5: datatset -> dataset
- line 44: a feature is functional important -> a feature is functionally important
- line 113: ")" missing in equation

**Limitations:**

Authors briefly discuss the lack of a ground truth to evaluate SAEs.

I think discussion of evaluation metrics could be a bit more divided (maybe into metrics to assess interpretability and metrics to assess how close to the original network a network with inserted SAEs is).

---

> ### Author Rebuttal · Authors · 2024-08-05
>
> Thanks for your comments and your typo spotting and notation improvement suggestions!
>
> ## Addressing main misunderstanding
> After reading your review, it appears we failed to convey the core benefit of our method. You state
> > the authors state that "locally trained SAEs are capturing information about dataset structure that is not maximally useful for explaining the algorithm implemented by the network." but do not show that their e2e SAEs are significantly more interpretable or helpful at explaining the algorithm implemented by the network, instead the e2e SAEs are merely "at least as interpretable as SAElocal"
>
> Indeed, we did not find that **individual** features found by e2e SAEs were more interpretable than local SAEs. Though we didn't expect them to be, nor was it the purpose of this work. The important contribution of this work is that in order to explain the activations at a particular point in the network, our method requires fewer than half of the features that other methods require. We illustrate this with the pareto curves in Figure 1 (left). This dramatically reduces the description length of any feature or circuit that uses SAEs. Shortness of description is an essential aspect of interpretability.
>
> In addition, we show that we need far fewer features in total to explain the network activations over the whole dataset (Figure 1 (right)).
>
> ## Response to Limitation
> > Authors briefly discuss the lack of a ground truth to evaluate SAEs. I think discussion of evaluation metrics could be a bit more divided (maybe into metrics to assess interpretability and metrics to assess how close to the original network a network with inserted SAEs is).
>
> We do have a section-level separation between both types of metrics (Sections 3.1-3.3 for the latter type and Section 3.4 for the former), and we link to the sections alongside a short explainer of them at the bottom of the introduction (lines 75-86). We’re open to alternative division if preferred.
>
> In addition, we have run more experiments in response to other reviews that evaluate the SAE setups on a set of downstream tasks.
>
> ## Other responses to Weaknesses and Questions
>
> > line 50: but we no longer optimize activation reconstruction. -> but you do use activation reconstruction for SAEe2e+ds, don't you?
>
> Yes. We train for downstream activation reconstruction SAEe2e+ds, but at all subsequent layers, rather than at the current layer. In this paragraph we are introducing SAEe2e, which does not involve any training for activation reconstruction.
>
> > Maybe I just missed i but where in the network do you plug in the SAEs? the residual stream/decoder layer output?
>
> All of our SAEs are inserted in the residual stream (lines 160). We'll add this to the main figure and table for further clarity.
>
> > You say that taking different computational pathways through subsequent layers of the network might be a problem, but do you have any evidence for this happening? is this just section 3.3.2?
>
> Primarily the evidence for this is in Section 3.2, especially in Figure 2, as well as in Section 3.3.2. Figure 2 directly shows the e2e SAE’s output activations taking a very different pathway through the network, as the activation reconstruction MSE is much higher and significantly increases downstream through the layers of the network, rather than converging. Section 3.3.2 provides additional evidence that the pathway through the network is under-constrained by the KL divergence loss of e2e SAEs alone. These problems are resolved by e2e+ds SAE training.
>
> > While reading, I was wondering if taking different computational pathways through subsequent layers of the network is still a problem if only one SAE is trained at a time. There should be little incentive for computing additional information since the original network post SAE insertion would not have capacity to handle these anyway, so it would only matter for the output directly.
>
> Even with one SAE trained at a time, it turns out that there is capacity for the original network to handle the SAE adding "new" features to the residual stream. Evidence for this can be seen in Figure 3b (middle), which shows that with two different seeds, a pure e2e SAE can learn a very different set of features (and still perform as well in the L0-CE loss pareto). While we're unsure how much of a problem this is, our e2e+ds formulation appears to resolve this issue at very little cost.
>
> > small things: line 68: not sure if it's necessary to mention goodhart's law since the target is not cross-entropy loss difference but interpretability so I would be more worried about this if you start optimizing interpretability measures
>
> This is reasonable, and is our argument too. Though it's worth mentioning that reviewer 2 and reviewer 5 both expressed concerns about Goodharting.
>
> >[notation improvement suggestions]
> line 5: datatset -> dataset.
> line 44: a feature is functional important -> a feature is functionally important.
> line 113: ")" missing in equation
>
> Thanks for spotting these errors and notation ideas! We'll incorporate these.
>
> ## Summary
> Thanks again for your comments and questions. We hope to have adequately clarified any uncertainties or misunderstandings w.r.t the core contributions of this work and other listed limitations or weaknesses. If so, we would ask you to reassess your review score, taking this response into account.

---

> > ### Comment · Reviewer_Hd7R · 2024-08-10
> >
> > Thanks for emphasizing again that your main contribution lies in reducing the description length. I do agree that this is a valid improvement over existing SAEs.
> > I'm sorry for being too unclear wrt my comment about evaluation metrics. You do discuss different metrics in different sections. I guess I was looking more for a high level picture. Sth like you want to maximize faithfulness between the network with and without inserted SAEs (to make sure you interpret the original network) and interpretability at the same time and how the different metrics serve these goals.
> > I do think it is hard to assess how much the reduction in description length helps with the goal of interpretability (maybe showing that pure reconstruction loss SAEs features can be misleading or sth could be helpful), so I did not update my score.

---

### Official Review · Reviewer_SeEL · 2024-07-02

**Soundness:** 3
**Presentation:** 2
**Contribution:** 2
**Rating:** 5
**Confidence:** 3

**Summary:**

The authors present their observations on modified sparse autoencoder (SAE) learning. In the proposed method SAE is trained to reconstruct original model weights (features). SAE is optimized with the KL-divergence loss between the model output and the output at the same location in the original model. An additional variant is training with the reconstruction loss computed after each layer between the original model and SAE output. The sparsity loss term is added to increase the number of weights that will converge to zero. The authors claim that their method explains the model features in a better way with fewer dictionary elements.

**Strengths:**

- Proposed a new training method for SAE with end-to-end loss that allows to use of fewer features from the original model
- The authors provided extensive analysis of their results
- The related work is sufficient
- The proposed method decreases the number of active dictionary elements twice while preserving the same CE loss decrease as a baseline SAE.

**Weaknesses:**

- The modification is very simple and raises the question of whether additional variants could work too, e.g. local SAE with MSE end-to-end loss (in addition to the layer reconstruction MSE)
- The numeric results in Figure 1 are very incremental: e.g. reduction of 0.05 in CE loss increase L0=100 is ~1.6% of the evaluation CE loss
- The interpretability is not well defined, in addition, the claim that fewer features means more interpretability should be proved by empirical evidence, white the authors showed that interpretability was not changed or changed significantly (appendix A7).  If I get it correctly, the main goal of the proposed method is interpretability but a few experiments were done to evaluate it. Also, qualitative results are not provided (anonymous url)
-  Std values are missing (for different random initialization seeds) in Figure 1
- The writing should be improved, Sec. 3.4 should be more self-contained
- It is not clear which method e2e or e2e-ds  is the best performer overall?

**Questions:**

- Can you please explain the diagram in Figure 1 why some blocks are of different shapes?
- It looks like that the Sec. 3.4 result is trivial, if we don’t encourage the SAE to reconstruct after wach activation, that will be the result.
- What is the number of parameters you train in SAE in total for all layers?
- Again, if the main benefit of the method is interpretability, I do not understand how this work addresses this property and what is its main contribution in improving interpretability.
- Are $W_e$ and $D$ the matrices defined for each layer? If they are - an appropriate indexing should be applied.

**Limitations:**

not discussed

---

> ### Author Rebuttal · Authors · 2024-08-05
>
> Thanks for you comments and suggestions
>
> ## Comments on Weaknesses
> > The interpretability is not well defined...
>
> The interpretability provided by an SAE contains two main components: The average number of SAE features required to interpret any particular model output (L0 - lower is better), and the individual interpretability of each of those features. Improving either one of these without hurting the other or the amount of model performance explained constitutes an improvement to interpretability by shortening the description length required to explain the model’s behavior. Shortness of description is an essential component of interpretability.
>
> As we show in Figure 1, e2e and e2e+ds SAEs are able to significantly reduce L0, requiring fewer active features compared to local SAEs explaining the same amount of model performance (as measured by the CE loss increase).
>
> While the interpretability of each feature is in general somewhat subjective and hard to measure, we put significant effort into gaining unbiased and rigorous measures by implementing automated interpretability scoring (Bills 2023).
>
> Overall, we believe the evidence is clear that our method produces SAEs which are significantly more useful for interpretability than SAEs produced by the baseline local training process.
>
> > The numeric results in Figure 1 are very incremental: e.g. reduction of 0.05 in CE loss increase L0=100 is ~1.6% of the evaluation CE loss.
>
> We believe that this is an inappropriate baseline. At the L0=100 position quoted, our method more than halves the CE loss increase, relative to the local SAE baseline. We would not call a halving of the error incremental. Recall that we are not trying to reduce the CE loss of the original model - merely to explain as much of the original model’s performance as possible.
>
> Second, although an increase of 0.05 in CE loss may seem relatively small, it corresponds to a much larger effective drop in model size or training FLOP via scaling laws between these quantities and loss. The drop in CE loss from the model learning more complex behaviors (which emerge at larger model sizes or training FLOP) generally reduces as the behaviors get more complex (relative to the drop in loss from the model learning simple bigrams or trigrams), and yet we are still especially interested in interpreting how the model performs these more complex behaviors.
>
> Finally, the quoted rises in CE loss are for one single SAE. in practice, an explanation of the model’s behavior will require SAEs in many layers throughout the model, so seemingly small errors can be multiplied in this way, leading to unfaithful descriptions (this is illustrated in Marks2024 (https://arxiv.org/abs/2403.19647)).
>
> > The modification is very simple and raises the question of whether additional variants could work too, e.g. local SAE with MSE end-to-end loss (in addition to the layer reconstruction MSE)
>
> Please see our response to reviewer TS92, where we agree that this would be an interesting variant to try, along with a long list of others.
>
> > Std values are missing (for different random initialization seeds) in Figure 1
>
> We provide some evidence in Figure 18 that the results of Figure 1 are robust to different random initialization seeds. Unfortunately, training enough SAEs to get robust estimates of the standard deviations for each datapoint would be prohibitively expensive.
>
> > It is not clear which method e2e or e2e-ds  is the best performer overall?
>
> We recommend e2e-ds as the best performer overall. It achieves a similar performance explained to e2e SAEs while maintaining activations that follow similar pathways through later layers compared to the original model (highlighted in section 3.2 and the conclusion).
>
> ## Responses to Questions
> > Can you please explain the diagram in Figure 1 why some blocks are of different shapes?
>
> The different shapes are to distinguish qualitatively different operations, such as the unembed (which projects up from the residual stream to the logits), and the SAE (which we train). These are labelled in the diagram.
>
> > It looks like that the Sec. 3.4 result is trivial, if we don’t encourage the SAE to reconstruct after wach activation, that will be the result.
>
> We believe you may have confused Section 3.4 with Section 3.3.2, or some other section. If so, Section 3.3.2 was not intended to be counterintuitive. If not, see our response to questions on evaluating interpretability above.
>
> > Are W_e and D the matrices defined for each layer? If they are - an appropriate indexing should be applied.
>
> We do only ever train one SAE at a time, but we will add some layer indices for clarity, thanks for bringing this to our attention.
>
> > What is the number of parameters you train in SAE in total for all layers?
>
> As mentioned on line 166, the number of dictionary elements in each SAE is fixed at 60 times the size of the GPT-2 small residual stream (60 x 768 = 46080 dictionary elements), so W_e and D both have sizes 46080 x 768, b_e is a 46080 dimensional vector, and b_d is a 768 dimensional vector, meaning that each SAE has just over 70 million parameters.
>
> ## Summary
> We thank you again for your comments and questions. If we have adequately addressed your concerns, we would kindly ask you to reassess your review score, taking this rebuttal into account.

---

> > ### Comment · Reviewer_SeEL · 2024-08-10
> >
> > Thank you for the rebuttal.
> >
> > In Figure 1 you present CE loss increase, not the error, so I don't understand you claim of "We would not call a halving of the error incremental.".
> >
> > I meant Sec. 3.2 not 3.4

---

> > > ### Author Response · Authors · 2024-08-12
> > >
> > > > In Figure 1 you present CE loss increase, not the error, so I don't understand you claim of "We would not call a halving of the error incremental.".
> > >
> > > Apologies if our use of the terms "error" and "CE loss increase" caused confusion in the rebuttal, we in fact did mean to use them interchangeably. To expand: The optimal CE Loss Increase (the metric used in Figure 1) is 0. Achieving this would mean that we could splice in the SAE, run the SAE activations through the rest of the network, and achieve exactly the same output as the original model. This would thus give us a complete representation of the features that matter for the output of the network.
> > >
> > > When we splice in a "vanilla" local SAE and then pass the SAE activations through the rest of the network, the CE loss of that network is worse than that of the original network. The yellow line in Figure 1 (left) shows how much worse it is. E.g. at an L0 of 100, it gives a CE Loss Increase of 0.11 compared to the original model.
> > >
> > > When we splice in an e2e SAE (either pure e2e or e2e+ds), we only get a CE loss increase of 0.05. So the amount of CE Loss Increase has ~halved from 0.11 to 0.05. As mentioned in our rebuttal, we do not think this is a small difference when considering that models containing many times more parameters only slightly decrease CE loss as per model scaling laws. Or perhaps more starkly, to get the same amount of CE Loss as our e2e models with L0=100 (CE Loss Increase of 0.05), a local SAE needs to activate ~400 features on average for each input (i.e. 4 times larger description length!).
> > >
> > > > I meant Sec. 3.2 not 3.4
> > >
> > > Ah, in this case we do agree that it is at least unsurprising that pure e2e performs much worse than local SAEs on this metric. Although we do find it valuable to know exactly how close our e2e+ds and local are on downstream layers. We use a reconstruction loss at downstream layers in the e2e+ds case, but not the local case, so it wasn't clear a priori how this plot would turn out for the optimal hyperparameters in each setting. It could have turned out that a smaller downstream reconstruction coefficient was optimal for e2e+ds SAEs, meaning that the downstream reconstruction MSE would be worse and we would be more concerned about the SAE taking different computational pathways through the network.
> > >
> > > We hope this helps clarify your remaining concerns.

---

> > > > ### Comment · Reviewer_SeEL · 2024-08-12
> > > >
> > > > Thank you for the explanations. I think that the paper is still borderline and should be improved. I raise my score appropriately.

---

### Official Review · Reviewer_TS92 · 2024-07-12

**Soundness:** 4
**Presentation:** 4
**Contribution:** 3
**Rating:** 6
**Confidence:** 4

**Summary:**

This paper proposes a way to train sparse autoencoders (SAEs) that encourages them to learn features that are causally relevant to the model's output. This is done by replacing the usual SAE reconstruction objective - an L2 penalty between original activations and their reconstructions - by the KL divergence between the model's output distribution and the distribution when activations are replaced by their reconstructions. Additional loss terms encourage the reconstructions to lead to downstream layer activations similar to the original downstream layer activations. The sparsity penalty on feature activations is kept as in the "vanilla" SAE.

The paper finds that this approach is a clear Pareto improvement over vanilla
SAEs w.r.t. the tradeoff between sparsity of the feature activations (L0) and
loss recovered when using reconstructions. An automatic interpretability pipeline is used to compare the features learned to vanilla SAE features. A statistically significant advantage for e2e+ds SAEs is found.

The paper also attempts to avoid a potential failure mode of the approach, whereby the SAE may learn to exploit different pathways through the model in order to drive down the KL divergence to the original output distribution, because this is easier from an optimization point of view (as the reconstructions are no longer optimized to match the original activations at the layer where the SAE is applied). This is why one of the SAE variants considered includes an L2 penalty for downstream layer activations too. Results here show that using only the KL penalty in the loss produces very different downstream activations compared to adding these additional terms, suggesting that the

**Strengths:**

- it is an important open problem whether SAEs trained solely using activations from some LLM layer will learn all causally important features for the LLM at this layer if trained without supervision from downstream activations. The paper makes some progress on this problem, suggesting that "vanilla" SAEs may struggle to learn such causally relevant features.
- the methodology is careful and often considers alternative hypotheses or potential pitfalls in the analyses.
- the paper is clearly written

**Weaknesses:**

- given that the KL divergence to the output distribution is incorporated in the
loss function, it is not that surprising that the methods in this paper have
better values of the loss recovered metric (to their credit, the authors
acknowledge this). To truly conclude the superiority of the suggested method for
surfacing *individual* causally important features, it would be extremely helpful to have some external (to the KL metric)
evaluation.
- To some extent the paper shows such evaluations. For example, in Appendix G.3 it is shown that vanilla SAEs represent well a feature that is not causally important, and the e2e SAEs in turn represent it poorly. However, what is *really* required to establish superiority is the opposite: to exhibit a causally important feature not represented by vanilla SAEs, but represented by e2e ones.
- the key problem when not using the same-layer activations as a reconstruction target for the SAE is (as the authors describe) that the optimization may prefer to learn features that achieve a good KL divergence value eventually, but do so through "pathological" pathways in the model. To fix this, the authors encourage closeness with activations starting from the next layer up. However, such "pathological" pathways may exist in a single layer (this was shown e.g. here https://arxiv.org/abs/2311.17030). So it is unclear whether the problem has been fully overcome or rather restricted to a more narrow part of the model. Furthermore, even if reconstructions of downstream layers are similar to the true activations of these layers, this does not establish that the individual features themselves are not "pathological". Again, some additional, *per-feature* evaluation is needed to make the (strong) claims of the paper more believable.
- out of the three kinds of SAEs considered - vanilla, e2e, e2e+ds - there seems to be a missing one implied by the others: an SAE that encourages faithful reconstructions only of the *next* layer activations, and does not involve the KL divergence loss. Would such an SAE lead to similar improvements? If so, this would be stronger evidence, as the KL divergence w.r.t. the final output distribution won't be a part of the loss.

**Questions:**

- did you try the "next layer only" SAE variant described in the weaknesses?
- how can we get a more fine-grained picture of how individual features change between the vanilla SAE and the e2e ones?

**Limitations:**

- I think the main limitation that I would have loved to see addressed more in the paper is that the evaluations are somewhat indirect w.r.t. the main claim of the paper. Even if in some average sense the e2e SAEs' reconstructions lead to better KL divergence, we still don't know how this plays out on the level of individual features.

---

> ### Author Rebuttal · Authors · 2024-08-05
>
> Thank you for engaging with our paper and giving well-considered feedback.
>
> Regarding the main limitation you’ve mentioned, we agree that it would be helpful for the paper’s narrative to have other metrics of functional importance than KLDiv, including evaluations on an individual feature level. However, there is an issue with this in practice, which relates to your comment
> > To some extent the paper shows such evaluations. For example, in Appendix G.3 it is shown that vanilla SAEs represent well a feature that is not causally important, and the e2e SAEs in turn represent it poorly. However, what is really required to establish superiority is the opposite: to exhibit a causally important feature not represented by vanilla SAEs, but represented by e2e ones.
>
> The extent to which features exist in one dictionary and do not exist in another is, unfortunately, not a binary concept in practice. We often find that there are directionally similar, but not identical features shared across dictionaries. Or there are clusters of features that are overrepresented in one dictionary and underrepresented (but not absent entirely) in another. Therefore demonstrating that “a causally important feature not represented by vanilla SAEs, but represented by e2e ones” may not reflect the metric that matters overall: the global functional importance of all the features.
>
> Nevertheless, we agree that it would be desirable to be able to identify compelling narratives about individual features across dictionaries and their improved functional importance. Unfortunately the differences are matters of degree – quantitative rather than qualitative, which makes qualitative stories more difficult to tell. We were able to identify one such qualitative story. However, despite efforts to identify qualitative stories in the direction suggested by the reviewer (“what is really required to establish superiority is the opposite: to exhibit a causally important feature not represented by vanilla SAEs, but represented by e2e ones.”), we found that the data did not suggest these qualitative narratives despite the quantitative differences being present on the global level.
>
> As for other external evaluations that don’t directly measure the CE loss over the same distribution used to train the SAE, we have run some additional experiments on a selection of downstream tasks. These are described in the Author Rebuttal at the top of this page
>
> > When not using same-layer reconstruction... it is unclear whether the problem has been fully overcome or rather restricted to a more narrow part of the model.
>
> We agree that our existing e2e+ds implementation does not eliminate the problem but rather restricts it to a narrower part of the model. Our initial concern with adding a reconstruction loss term at the current layer was that it would reconstruct too many non-functional features present in the residual stream at the current layer. Though we have come to agree that the difference between reconstructing the current layer and a subsequent layer is merely quantitative rather than qualitative. As mentioned in the "Other variations of e2e+ds" section in the Author Rebuttal, we unfortunately have not run the extensive sweeps on the e2e+ds variations needed to give an insight to this question empirically, so it is possible that additionally reconstructing the current layer's activations would perform just as well on the L0-CE Loss Pareto while further restricting the computational pathways available to the model.
>
> Regarding the question
> > did you try the "next layer only" SAE variant described in the weaknesses?
>
> please see our comment on this in our Author Rebuttal at the top of the page.

---

> > ### Comment · Reviewer_TS92 · 2024-08-10
> > **Valuable, if negative, results**
> >
> > Thank you for the detailed and thoughtful rebuttal, as well as the overall response.
> >
> > I appreciate the additional evaluations of your proposed method.
> > My interpretation of the additional experiments is that they provide at best mixed evidence
> > for the superiority of the proposed SAE variants. That being said, I appreciate the honesty
> > of the authors, and I believe these results will be valuable (and perhaps surprising) to the
> > interpretability community, so I am in favor of disseminating these results widely.
> >
> > This is why I maintain my recommendation to accept this work (though I will not be changing
> > my score at present).

---

### Official Review · Reviewer_byuk · 2024-07-12

**Soundness:** 3
**Presentation:** 3
**Contribution:** 3
**Rating:** 6
**Confidence:** 3

**Summary:**

The paper proposes new framework to improve the interpretability of large language models. Inspired by the work proposed in Anthropic's "", the proposed work replaces the original reconstruction loss in the sparse auto-encoder (SAE) with KL-div loss, and additionally add further constraint to minimize the error between reconstructed downstream activations and original activations in the LLM. Empirical results demonstrates the adavantage of the proposed new SAE_{e2e} architecture series.

**Strengths:**

S1: The paper considers new formulation to further improve the interpretability of the large language model.

**Weaknesses:**

W1: The authors may further explain the intuition between the evaluation metrics. The current version of the evaluations is not explicitly clear to me why the proposed method is advantageous in comparison to the baselines.

W2: It is now clear how the KL divergence is implemented to replace the reconstruction loss. KL is only practically useful when the distributions are available. However, in this problem, it is unclear how the distribution loss between the two deterministic feature outputs are computed.  The motivation of the replacement between KL divergence and the L2 reconstruction is also not very clear to me, and these two losses should be ultimately equivalent and optimal at the same parameters global optima, although the loss changes the optimization landscapes.

**Questions:**

I appreciate it if the authors could please see the above weakness for my questions.

**Limitations:**

Yes the authors have adequately discussed the limitations of the work.

---

> ### Author Rebuttal · Authors · 2024-08-05
>
> We thank the reviewer for their comments and questions. We believe there may be a core misunderstand of our method which we clarify in the response to Weakness 2.
>
> ## Response to Weaknesses
> ### Weakness 1
> > W1: The authors may further explain the intuition between the evaluation metrics. The current version of the evaluations is not explicitly clear to me why the proposed method is advantageous in comparison to the baselines.
>
>
> In the paper we present three main components to evaluating the performance of an SAE. Below we give some more intuition behind them.
> 1. If we are to sum up dictionary elements of our SAE and use those in place of the real model activations, how many dictionary elements do we need to use to get similar performance to the real model? Figure 1 (Left) shows that we require fewer dictionary elements on average (L0) to explain the same amount of CE loss degradation (which is the difference between the CE loss of the original model and the model when splicing in SAE features).
> 2. In Figure 1 (right), we measure the total number of dictionary elements needed in the SAE to explain the network’s behavior over the whole dataset. While less important than the metric above, having fewer dictionary elements in total means that there are fewer variables needed to describe the behavior of the network on your distribution. Note that “alive dictionary elements” is the effective measure of the number of variables needed, as some dictionary elements become forever inactive during training (it’s worth noting that the recent work of Gao2024 (https://cdn.openai.com/papers/sparse-autoencoders.pdf) has reduced the killing of dictionary elements with better initialization and an augmented loss term, though we don’t have a reason to expect that employing these techniques will change the structure of the pareto curves presented in our figure.
> 3. We want to know whether we can understand each SAE dictionary element individually. We implemented automated interpretability scoring, which provides a scalable and unbiased quantitative measure of interpretability. For an introduction to this technique, see  Bills et al. [2023] https://openaipublic.blob.core.windows.net/neuron-explainer/paper/index.html. In short, to score the interpretability of a single SAE feature, GPT-4-turbo is prompted to write a short english-language explanation of the feature, then GPT-3.5-turbo is prompted to try to predict the specific activations of the SAE feature on each token, as we explain in section 3.4. From the automated interpretability scoring, we find strong evidence that the features in our e2e+ds SAEs are not less interpretable than the features in the local SAEs at a similar amount of model performance explained. We even find that they are significantly more interpretable in some layers, such as in layers 2 (p = 0.0053) and 6 (p = 0.0005).
>
> It may be useful to note that several other works in the area have used the metric outlined in point 1 to evaluate SAEs e.g. Figure 5 in Bills2024 (https://cdn.openai.com/papers/sparse-autoencoders.pdf), Rajamanoharan2024 (https://arxiv.org/abs/2404.16014)(note that this uses a “loss recovered” term which is calculated from the average CE loss of the model when splicing in the SAE), Kissane2024 (https://arxiv.org/abs/2406.17759). Additionally, many works have used automated interpretability to evaluate the interpretability of SAE dictionary elements (e.g. Cunningham2023 (https://arxiv.org/abs/2309.08600), Templeton2024 (https://transformer-circuits.pub/2024/scaling-monosemanticity/),
>
> In addition, in the Author Rebuttal at the top, we've run additional evaluations. The results measure how much task performance degradation there is when SAE dictionary elements are spliced into the model on some of the subtasks used in Marks2024 (https://arxiv.org/abs/2403.19647).
>
>
> ### Weakness 2
> > However, in this problem, it is unclear how the distribution loss between the two deterministic feature outputs are computed.
>
> The KL divergence is computed for the output of the LLM. This output is a probability distribution over next tokens, motivating our use of KL divergence.
>
> > The motivation of the replacement between KL divergence and the L2 reconstruction is also not very clear to me, and these two losses should be ultimately equivalent and optimal at the same parameters global optima, although the loss changes the optimization landscapes.
>
> For an SAE with perfect reconstruction, both L2 and KL loss would be zero. However, the sparsity_loss and reconstruction_loss terms trade off against one another (given a fixed dictionary size). This means that the global optimum does not result in an SAE with perfect reconstruction.
>
> We use KL divergence rather than L2 reconstruction at the output as the outputs are probability distributions (and the original network was trained with the standard language modeling loss which is the KL divergence between its outputs and the one-hot labels). For SAE_local and SAE_e2e+ds, we have a loss term at intermediate layer(s) in the network. Here, we use the L2 reconstruction as these activations are not probability distributions.
>
> ## Summary
> We thank the reviewer again for their comments and questions, and kindly ask that they reassess the review score if their concerns were adequately addressed by this response.

---

> > ### Comment · Reviewer_byuk · 2024-08-12
> > **Thanks for your rebuttal**
> >
> > Thanks for the rebuttal and the further clarifications on my questions. Given the explanations on the motivations behind using KL to replace L2, I decide to increase my score. I also encourage the authors to include the discussions on these differentiating factors between L2 and KL into the final version of their paper.

---

### Author Rebuttal · Authors · 2024-08-06

We'd like to thank the reviewers and ACs for their time. We're delighted to hear that our work "makes some progress on ... an important open problem" (R2), "provides extensive analysis of their results" (R3), "provide an intuitive solution" (R4), introduces "exciting techniques" and has some "extremely interesting" analysis (R5). We're also glad to see that the reviewers who gave the highest ratings also had the highest confidence in their ratings (R2, R5).

Responses to individual comments/questions are provided alongside each review. Most notably, we hope that we've cleared up what we believe to be a core misunderstandings of the paper's contribution for R1 and R4. Below, we respond to concerns that were shared across multiple reviewers.

## Other variations of e2e+ds
Reviewer TS9212 (R2) and reviewer SeEL02 (R5)  commented about trying other variations of the e2e+ds SAE. Reviewer TS9212 asks _“did you try the "next layer only" SAE variant described in the weaknesses?”_. Reviewer SeEL02 states _“The modification is very simple and raises the question of whether additional variants could work too, e.g. local SAE with MSE end-to-end loss”_.

We agree that these would both be interesting variants to try, although there are also many other variants omitted in our experiments. A more complete set of variants might be:

- *Our local
- Local + next layer reconstruction
- Local + arbitrary downstream layer reconstruction
- Local + multiple arbitrary downstream layer reconstruction
- Next layer reconstruction only
- Arbitrary downstream layer(s) reconstruction only
- Multiple arbitrary downstream layer reconstruction only
- Local + KL Div
- Local + next layer reconstruction + KL Div
- Local + arbitrary downstream layer reconstruction + KL Div
- Local + multiple arbitrary downstream layer reconstruction + KL Div
- Next layer reconstruction + KL Div
- Arbitrary downstream layer(s) reconstruction + KL Div
- *Multiple arbitrary downstream layer reconstruction + KL Div (i.e. our e2e+ds)
- *KL Divergence only (i.e. our e2e)

We're also interested in these variants, not least because it would be interesting to know if we can get much of the benefit of e2e+ds merely by reconstructing a more ‘local’ set of layers and avoiding the backpropagation through most of the model. We expect that these variants will not greatly change the L0-CE Loss Pareto frontier, though it is likely that they would result in different properties w.r.t the computational pathway through the network. Since we could not focus on every variant, we chose to focus on e2e and e2e+ds since we believe that these were sufficient illustrations of our core thesis.

## Further evaluations

Reviewer TS9212 (R2) and reviewer ErCK12 (R5) had concerns that the paper did not provide evaluation metrics that were not biased by the KL divergence loss metric we introduced. Reviewer TS9212 states _“I think the main limitation that I would have loved to see addressed more in the paper is that the evaluations are somewhat indirect w.r.t. the main claim of the paper. Even if in some average sense the e2e SAEs' reconstructions lead to better KL divergence, we still don't know how this plays out on the level of individual features.”_ Reviewer ErCK12 states _“...it would greatly improve the paper if there were additional metrics for downstream SAE quality that were tested (e.g. some of the methods described in section 4.4).”_

We think these are good criticisms of the paper. For this reason, we’ve run additional evaluations on specific tasks. The experiments are described below, and some results can be seen in the attached pdf.

### Methodology
We adapt the datasets from Marks et al. (2024), originally presented in Finlayson et al. (2021). This includes 4 variations of subject-verb agreement template data:
* Simple (The parent/s is/are)
* Within RC (The athlete that the manager/managers likes/like)
* Across RC (The athlete/athletes that the managers like do/does)
* Across PP (The secretary/secretaries near the cars has/have)

For our metric $m$ we take the difference in logits between the correct completion, and the same verb but swapped between singular and plural forms For example, given $x_\text{clean}=\texttt{The teachers}$, $m(x_\text{clean})$ will be $\text{logit}(\texttt{ are}) - \text{logit}(\texttt{ is})$.

Then the $\textit{faithfulness}$ of a network under an intervention is defined as $\frac{E_x[m(x_\text{clean} | \text{intervention})]}{E_x[m(x_\text{clean})]}$

Note that the fully zero or mean ablated model will have an average logit difference of zero, due to the construction of the dataset.

### Faithfulness of models with SAEs
We first test the faithfulness of the models with the SAEs inserted (Table 1 in pdf). All SAEs preserve most of the logit-difference, however there is significant variation across SAE types, layers, and tasks. The local SAEs in layer 10 have the worst faithfulness, although our e2e + ds SAE in layer 6 also has poor faithfulness across participial phrases.

### Number of nodes needed to explain behavior
We order SAE features in each Similar $L_0$ SAE by indirect effect (mean ablating one SAE feature at a time and measuring how the metric changes). We then measure the metric while preserving only the top k most important features (for $k \in [1, \cdots, 1200]$). The results are in Figure 1 of the pdf.

### Result Summary
While we see some (SAE type, SAE layer) combinations perform better than others, we see no clear patterns between SAE types over all the tasks. These results indicates that e2e SAEs do not provide an obvious benefit on these specific tasks. Even though we've shown that e2e SAEs are beneficial on the language modeling task over the full openwebtext distribution, further work would be needed to find specific tasks and subdistributions in which they provide the most benefit.

## Rebuttal Summary
We hope that our additional experiments and comments to the reviewers questions satisfy their concerns.

---

### Decision · Program_Chairs · 2024-09-25

**Decision:**

Accept (poster)

**Comment:**

**Summary of the paper:**
The paper proposes a new way to train sparse auto-encoders (SAE) to be used for interpreting pre-trained deep networks. Instead of directly minimizing the discrepancy of the feature out of SAE with that of an intermediate layer of the original network, it minimizes the discrepancy at the output layer, i.e., probability distribution over the next token in the case of LLMs considered in this work. It obtains the output for SAE by passing the SAE’s encoded feature through the rest of the layers of the original network. It also uses a sparsity inducing regularization. It further adds a reconstruction loss to some later layers.

**Summary of the reviews:**
The reviewers found the proposed method well-justified, the reasoning of the paper convincing, the limitations well-covered, the results mostly suggesting improved performance and the geometric study of the learnt features informative .

On the other hand, they found evaluating metrics not easy to interpret, not entirely convincing, or sometimes incrementally improved, the mitigation of inconsistent pathways still possibly problematic when moving the reconstruction to a later layer, a lack of ablation study when only the later reconstruction is used (without KLD on output).

**Summary of the rebuttal and discussions:**
The authors provided new experiments on a different evaluation metric than CE for faithfulness.
This along with the textual clarifications on the evaluation metrics rectified some of the concerns of the reviewers. However, some reviewers remained somewhat concerned about the metrics. Despite that, all reviewers are leaning towards acceptance.

**Consolidation report:**
As reviewers have pointed out a crucial strength of the paper is the outright transparency on the limitation of their approach and importantly the limitations of the experiments and evaluations in supporting their basic claims and, in fact, the paper often discusses alternative hypotheses. This mitigates most of the issues raised above, which are especially valid regarding the metrics. This strength in tandem with the fact that the paper has taken clear steps forward for SAE interpretation of LLMs, make a good paper.

**Recommendation:**
The AC agrees with the general tendency of all reviewers and recommends acceptance. The discussions (including the concerns raised by reviewers) and additional points and results provided by the authors during the rebuttal would be quite enlightening for future readers and thus the authors are encouraged to incorporate them in the camera ready version of the paper. It would be good to have a section discussing the metrics used and their limitations as was extensively discussed with the reviewers.